

# Evaluating multi-year, multi-site data on the energy balance closure of eddy-covariance flux measurements at cropland sites in southwest Germany

Ravshan Eshonkulov[1,3], Arne Poyda[1], Joachim Ingwersen[1], Hans-Dieter Wizemann[2], Tobias KD Weber[1], Pascal Kremer[1], Petra Högy[4], Alim Pulatov[5] and Thilo Streck[1]

[1]Institute of Soil Science and Land Evaluation, Biogeophysics, Hohenheim University, Emil-Wolff Str. 27, 70593 Stuttgart, Germany

[2]Institute of Physics and Meteorology, Physics and Meteorology, Hohenheim University, GarbenStr. 30, 70593 Stuttgart, Germany

[3]Environmental Protection and Ecology, Karshi Engineering Economic Institute, Mustakillik Avenue 225, 180100 Karshi, Uzbekistan

[4]Institute of Landscape and Plant Ecology, Plant Ecology and Ecotoxicology, Hohenheim University, August-von-Hartmann-Str. 3, 70593, Stuttgart, Germany

[5]EcoGIS center, Tashkent Institute of Irrigation and Agricultural Mechanization Engineers, Kary Niyoziy Str.39, 100000 Tashkent, Uzbekistan

*Correspondence to*: Eshonkulov Ravshan (ravshan.eshonkulov@uni-hohenheim.de)

**Abstract.** The energy balance of eddy covariance (EC) measurements is typically not closed. This so-called energy balance closure (EBC) problem has been a long-standing issue in micrometeorology. It is a main challenge in evaluating and interpreting EC flux data. EBC is crucial for validating and improving regional and global climate models. To investigate the reasons behind EBC more closely for agro-ecosystems, we analysed EC measurements from two climatically contrasting regions (Kraichgau (KR) and Swabian Jura (SJ)) in southwest Germany. Data were taken at six fully equipped EC sites from 2010 to 2017. The gap in EBC was quantified by ordinary linear regression, by the energy balance ratio (EBR) between turbulent fluxes and available energy, and by the residual energy term. In order to examine potential reasons for differences in EBC, we compared the EBC under varying environmental conditions and investigated a wide range of possible controls. Statistical analyses were conducted for the whole data set to test the effects of different regions, years, sites and crops on EBC. We also investigated whether EBC was a function of buoyancy, friction velocity, or atmospheric stability. The time-variable footprints of all EC stations were estimated based on data measured in 2015, complemented by micro-topographic analyses along the prevailing wind direction. The lowest mean annual energy balance gap was 17 % in KR and 13 % in SJ. Highest EBRs were commonly measured for winds originating from the prevailing wind direction. The variation of EBC was higher in winter than in summer. The measurement site exerted a statistically significant effect on EBC, but not crop or region. The spread of EBR distinctly narrowed under unstable atmospheric conditions, strong buoyancy, and high friction velocities. Smaller footprint areas led to better EBC due to increasing homogeneity. Flow distortions of winds that first travelled past the back head of the anemometer affected EBC negatively.



# 1 Introduction

Studying turbulent exchange at the land surface is important for assessing water cycling, plant growth and carbon fluxes of ecosystems and for enhancing soil-crop, climate and weather models. The currently best technique to measure these fluxes is the eddy covariance (EC) method. The EC method allows a direct and the most accurate measurement of turbulent fluxes in
the soil-plant-atmosphere system (Baldocchi et al., 2001; Burba, 2013). In EC flux data, the measured available energy (incoming net radiation minus ground heat flux) is generally higher than the sum of turbulent exchange fluxes (latent plus sensible heat). Accordingly, either the turbulent fluxes are incompletely captured or the measured available energy is positively biased. Thereby, the lack of energy balance closure (EBC) has been a long-standing problem in EC measurements and one of the most frequently discussed concerns in micrometeorological research (Foken, 2008a).

Across the world, research is being conducted to understand the reasons for the energy imbalance. One of the most extensive global EC networks is FLUXNET, with more than 500 EC towers around the world (Wilson et al., 2002). AmeriFlux is operating in North, Central and South America to measure ecosystem $CO_2$, water, and energy fluxes (Peng et al., 2017). In 2000, an Energy Balance Experiment (EBEX) was conducted to determine the reasons for the energy imbalance of EC measurements over irrigated cotton fields. The results showed that the net radiation differed by up to 10 W m$^{-2}$ across a
single field (Kohsiek et al., 2007). In a review (Foken, 2008b) of EBC, the most important factors for the energy imbalance were summarized as measurement errors of the energy balance components, incorrect sensor configurations, influences of heterogeneous canopy height, unconsidered energy storage terms in the soil-plant-atmosphere system, inadequate time averaging intervals, and long-wave eddies (mesoscalic circulations) (Foken, 2008b; Jacobs et al., 2008; Wilson et al., 2002). The influence of site characteristics (vegetation type, canopy height, terrain, etc.) on the EBC has been studied extensively.
Wilson et al. (2002) reported no clear differences in EBC between flat and sloped terrain sites across 22 research sites. A comparison of two different agroecosystems in China, a degraded grassland and a maize cropland, showed a similar EBC of about 80 % (Du et al., 2014). The comparison of a mature boreal jack pine forest and a jack pine clear-cut site by Kidston et al. (2010) revealed that, depending on the surface characteristics, the loss of low frequencies can contribute significantly to the energy imbalance. The impact of canopy height on EBC has been investigated at various locations around the world.
Wilson et al. (2002) studied the relationship between vegetation height and EBC and concluded that there was no apparent correlation. However, considering stored energy in the soil-plant-atmosphere zone noticeably improved EBC (Jacobs et al., 2008; Meyers and Hollinger, 2004; Zeri and Sá, 2010). Meyers and Hollinger (2004) compared the energy stored or released by $CO_2$ exchange and crop enthalpy change of a maize and a soybean canopy. Maize crops stored more energy than soybean crops. Eshonkulov et al. (2018) reported that mean EBC was improved from 78 % to 87 % when minor energy storage and
flux terms were taken into account during the main vegetation period.

The last decade has been marked by an increased interest in identifying the contributing source area (footprint) and evaluating the representativeness of the EC flux data for the field of interest (Göckede et al., 2006; Kljun et al., 2004; Schmid, 2002). Knowledge about the footprint is important to clarify whether the EC station measures local or nonlocal





energy fluxes (Eugster and Merbold, 2015; Pirk et al., 2017). The mismatch of measurement scales is also considered to be one reason for the energy imbalance (Sánchez et al., 2010; Xu et al., 2017). Currently, a variety of models is used to estimate footprint areas. Most analytical footprint models assume a homogeneous flux source area. Footprint calculation for heterogeneous sites requires much greater computational effort. Detailed surface characteristics must be included (Mauder et

al., 2013). Despite the existing methods and studies, Stoy et al. (2013) concluded that the relationship between footprint and EBC in agricultural cropland has not been sufficiently studied.

Currently, a widely discussed problem in assessing the EBC of EC measurements is related to the time averaging period. The main point is that an energy transport with near-surface secondary circulations (large eddies) cannot be measured with a single EC station (Cava et al., 2008; Foken, 2008b; Xu et al., 2017). Kidston et al. (2010) found that the EBC at a forest site

peaked at 90 % at a 240 min averaging interval. At a boreal forest site, Sánchez et al. (2010) applied different time averaging intervals and found that increasing the interval from the traditional 30 min to one day improved the EBC from 75 to 100 %. At an irrigated cotton field, however, Oncley et al. (2007) found that extending the time averaging interval up to 4 h did not result in a higher contribution of turbulent fluxes, and that the contribution of low turbulent fluxes was less than 10 W m$^{-2}$. In most cases, the 30-min standard averaging period has proven to be the best compromise to simultaneously capture most of

the turbulent fluxes while fulfilling the precondition of stationarity (Charuchittipan et al., 2014; Masseroni et al., 2014; Sun et al., 2006).

The present study evaluates the energy balance at croplands. The analysis is based on EC measurements performed from 2010 to 2017 at six fully equipped sites in two climatically different regions of southwest Germany. We hypothesized that multi-year, multi-site observations will provide new insights into the nature of the energy imbalance of EC flux

measurements. The main objectives of this study are to evaluate the EBC over agricultural crop rotations and to examine the dependency of the EBC on crop type, site characteristics, wind direction, atmospheric conditions, and footprint.

## 2. Materials and Methods

### 2.1 Site description

The study sites in the Kraichgau region (KR) are located at "Katharinentalerhof", approximately 4 km north of the city

Pforzheim (48.92° N, 8.70° E). The three adjacent fields are called EC1, EC2, and EC3, with sizes of 14.9, 23.6 and 15.8 ha, respectively (Fig. 1). The experimental fields are mostly flat. The prevailing wind direction is west. Southwards about 500 m away from the EC stations marks the site of the former waste dump of Pforzheim. Its elevation is about 41 m higher than the surroundings. KR is one of the warmest regions in Germany. Mean temperature and annual precipitation were 9.4 °C and 889 mm during 1981−2010 (Meteorological station Pforzheim-Ispringen, German Weather Service, about 3 km from the

research sites). The soils of this region developed from deep loess layers. The underlying rock is shell limestone. Detailed



information about meteorological and soil conditions can be found in Imukova et al. (2016), Ingwersen et al. (2015), Wizemann et al. (2014) and also in Table 1.

The Swabian Jura region (SJ) is characterized by a colder and harsher climate than KR due to its higher elevation. Accordingly, crops are generally sowed and harvested later than in KR. The prevailing wind direction is southwest to west.

Mean temperature and annual precipitation were 7.5 °C and 1042 mm during 1981−2010 (Meteorological station Merklingen, German Weather Service, about 2 km from the research sites). Information about meteorological and soil variables is given in Table 1. SJ is the largest contiguous karst landscape in Germany, with generally rather shallow soils. The study sites are located close to the city of Merklingen (Fig. 1). The sizes of EC4, EC5, and EC6 are 8.7, 16.7 and 13.4 ha, respectively. While EC4 and EC5 adjoin, EC6 is situated 1.5 km north of the two other sites (Fig.1).

The most frequent crops grown at the six sites were winter wheat and silage maize (Table 2). Crop rotation in SJ was more diverse than in KR. In 50 % of the site years in KR, winter wheat was grown; this value was only 25 % in SJ. Furthermore, only three different crops were grown at KR stations and six in SJ. At all sites, farmers frequently grew cover crops between winter and summer crops. These were mainly mustard, phacelia or multi-species mixtures.

## 2.2 Eddy covariance (EC) measurements

EC stations were installed at the center of each field in spring 2009 (Ingwersen et al., 2011; Wizemann et al., 2014). Stations were equipped with a fast-response $CO_2/H_2O$ open-path infrared analyzer (LI-7500; LI-COR Biosciences, Lincoln, NE, USA) and a three-axis ultrasonic anemometer (CSAT3; Campbell Scientific Inc., Logan, UT, USA). In KR, in early 2009 the CSAT orientation at EC1 and EC3 was 230°, at EC2 255°. In late April 2010, the orientation was changed to 170° and varied

over the subsequent years between 160° and 190°, ensuring that winds from the prevailing wind direction (west and east) enter the anemometer from the side. In SJ, the mean CSAT orientation was 220°±15 from late March 2010 until the end of 2017. The gas analyzers were factory-calibrated biannually. Sensor heights were adjusted to the increasing canopy heights, particularly during the vegetation periods of maize, leading to a distance of roughly 2–3 m between sensors and canopy. Among all EC stations, maximal sensor heights in KR and SJ were 6.00 m at EC2 (2014) and 4.80 m at EC6 (2010). The

minimal sensor height was roughly 2 m in both regions. All EC systems were powered by two 12 V batteries (each 240 Ah) charged by four 20 W solar panels. To enable continuous EC measurements during winter, a portable fuel cell system (Efoy Pro 800 Duo, FSC Energy AG, Brunnthal-Nord, Germany) was installed in autumn 2015 at EC2 and EC6. At the others stations the LI-7500 was shut down during the winter, mostly from late November to mid-March. The raw data of the gas analyzer and sonic anemometer were recorded at 10 Hz and stored on a CR3000 data logger (Campbell Scientific Inc.,

Logan, UT, USA).

Net radiation was measured using a 4-component radiometer (NR01, Hukseflux Thermal Sensors, Delft, The Netherlands). The radiometers were placed close to the EC stations, above the cropped field area, roughly 2 m above ground. Air





temperature and relative humidity were measured at a height of 2 m at each EC station using a temperature and relative humidity probe (HMP45C, Vaisala Inc, Helsinki, Finland). Soil temperature was measured at the depths of 0.02, 0.06, 0.15, 0.30 and 0.45 m (107 Thermistor probe, Campbell Scientific Inc., Logan, UT, UK). To measure the soil heat flux near the EC stations, three heat flux plates (HFP01, Hukseflux Thermal sensors, Delft, The Netherlands) were installed at a depth of

0.08 m. The soil volumetric water content at 0.05, 0.15, 0.30, 0.45 and 0.75 m depth was monitored with frequency-domain reflectometry (FDR) sensors (CS616, Campbell Scientific Inc., Logan, UT, USA). In the shallow soil at EC6, however, soil variables could be measured only down to 0.3 m. Data from thermistor (0.02 m and 0.06 m) and FDR sensors (0.05 m) were used to calculate the soil heat storage between the soil heat flux plates and the ground surface. Precipitation was measured with a 0.2 mm tipping bucket rain gauge (ARG 100, Environmental measurements Ltd., North Shields, UK) which was

installed 1 m above ground. The rain gauges were recalibrated once per year.

### 2.2.1 Post-field data processing and quality control

High-frequency raw data from 2010 to 2017 were processed with an averaging interval of 30 min using the software package TK3.1 (Mauder et al., 2013). The following settings were used to compute latent (*LE*) and sensible heat flux (*H*): spike

detection  (Vickers & Mahrt, 1997), planar fit coordinate rotation (Wilczak et al., 2001), correction of spectral loss (Moore, 1986), sonic virtual temperature conversion into actual temperature (Schotanus et al., 1983) and correction for density fluctuations (Webb et al., 1980). Additionally, the raw data of 2015 were processed with the software Eddypro® (Version 6.2.1, LI-COR Inc., 2012) to obtain input parameters (Obukhov length, standard deviation of lateral velocity fluctuations after rotation, friction velocity, mean wind speed and direction) for deriving flux source area (footprint). Data processing and

correction in EddyPro® was conducted with the same settings as in TK 3.1. Both programs yield comparable results (Fratini and Mauder, 2014).

As the quality criterion we used the nine flag system after Foken et al. (2004), provided by TK3.1. For evaluation, we used only data with quality flags 1–3 as suggested by Mauder and Foken (2011). Moderate (flags 4–6) and poor quality (flags 7–9) data were discarded. In a second step, a median filter was applied for additional de-spiking of half-hourly fluxes. The

filter removes all fluxes exceeding 5 times the median of the previous three days (Demyan et al., 2016). No gap-filling was performed in this study. Only high-quality measurements were used in the data analysis.

### 2.3 Energy balance closure of eddy covariance measurements

In the ideal case, the surface energy balance obeys the following equation:

$R_n - G = LE + H$                                                                      (1)




where $R_n$ is the incoming net radiation, $LE$ is the latent heat flux, $H$ is the sensible heat flux (both positive upwards) and $G$ is the ground heat flux (positive downwards). All components are expressed in W m$^{-2}$. Note that in Eq. 1 minor flux terms such as energy storage in the canopy or energy conversion by photosynthesis are neglected. All filtered half-hourly flux data for which all four components were available were used to calculate the EBC. Three measures were used to evaluate the EBC at

5   our sites. Firstly, the slope and intercept from ordinary linear regression (*OLR*) of turbulent fluxes ($H + LE$) against available energy ($R_n - G$) was assessed. In the ideal case of a fully closed energy balance, the slope and intercept of the linear regression are equal to one and zero, respectively (Ping et al., 2011; Wilson et al., 2002). In this study, we considered also the intercept (W m$^{-2}$) of OLR  in evaluating EBC as suggested by Franssen et al. (2010).

Secondly, the energy balance ratio (*EBR*) was calculated by:

$$EBR = \frac{H+LE}{R_n-G}, \qquad (2)$$

The third measure to assess the EBC of EC measurements in terms of absolute values is the computation of the energy balance residual (*Res*, W m$^{-2}$):

$$Res = R_n - G - H - LE, \qquad (3)$$

**2.4 Atmospheric conditions**

As a proxy for the role of shear and buoyancy in the production or consumption of turbulent kinetic energy, we used the friction velocity, $u^*$ (m s$^{-1}$), and the kinematic virtual temperature flux, respectively. The latter is the covariance ($w'T_v'$)

20   between vertical wind speed ($w$) and virtual temperature ($T_v$). As the virtual temperature can be replaced by the sonic temperature ($T_s$) with negligible loss of accuracy (Kaimal and Gaynor, 1991), we computed the virtual temperature flux from the covariance ($w'T_s'$) between $w$ and $T_s$.

The relationship between atmospheric stability and the EBC was examined using the atmospheric stability parameter as defined  by Stull (1988):

$$\zeta = z_m/L, \qquad (4)$$

where $z_m$ (m) is the measurement height of the sonic anemometer and $L$ (m) is the Obukhov length. The stability parameter expresses the relative roles of shear and buoyancy. Using $\zeta$, the stability of the atmosphere can be divided into three classes

30   (Franssen et al., 2010): stable ($\zeta \geq 0.1$), neutral ($-0.1 < \zeta < 0.1$) and unstable ($\zeta \leq -0.1$).



### 2.5 Footprint analyses and micro-topography

To determine the relationship between the contributing source area of turbulent fluxes and the EBC, we performed footprint analyses. We used the flux footprint prediction (FFP) online tool of a simple two-dimensional parametrization presented by Kljun et al. (2015) (http://geography.swansea.ac.uk/nkljun/ffp/www/). The footprint parametrization uses the Lagrangian

stochastic particle dispersion model (Kljun et al., 2002). As input parameters to the model, we used measurement height above ground, $z_m$ (m), displacement height, $z_d$ (m), mean wind speed (m s$^{-1}$), Obukhov length (m), standard deviation of horizontal wind speed (m s$^{-1}$), friction velocity, $u^*$, and wind direction (°). The measurement height ($z_m$) was calculated by eq. 5. The displacement height was calculated as in the EddyPro® software (LI-COR Inc., 2012) by eq.6:

$$z_m = z_{receptor} - z_d, \tag{5}$$

where $z_{receptor}$ is the instalment height of the sonic anemometer and the gas analyser, and $z_d$ is displacement height.

$$z_d = 0.67 * z_{can}, \tag{6}$$

where $z_{can}$ (m) is the time variable canopy height. Data for footprint analyses were constrained to u* > 0.1 m s$^{-1}$ and ζ ≥ −15.5.

Additionally, the micro-topography of the EC sites was determined along a transect in the prevailing wind direction (Fig. 1). About every two meters, the elevation of the fields above mean sea level was measured with a differential global positioning

system (DGPS) (Altus APS 3M, Septentrio, Belgium).

### 2.6 Statistical analyses

For the statistical analyses, we used all available data on energy fluxes from the onset of measurements (late March/early April) until harvest. In the case of maize, however, full data for the calculation of energy balances was generally available

from May. Autocorrelation of the data was tested using the Durbin-Watson test (Faraway, 2014). Analyses of variance (ANOVA) were used to test for significant effects of region, site, year and crops on EBC. Therefore, linear mixed models were defined (Piepho et al., 2004). The data were assumed to be normally distributed but heteroscedastic due to the different years. We based these assumptions on graphical residual analyses. Generally, the factors of interest were defined as fixed and interaction terms were considered. Remaining factors not included in the ANOVA were defined as random. Multiple

contrast tests (Bretz et al., 2011) were performed to identify significant differences between the different factor levels. These procedures were performed with the statistical software R (R Core Team, 2014) using packages *"multcomp"* for

simultaneous tests of linear mixed models (Hothorn et al., 2017), "*nlme*" for fitting and comparing the models (Pinheiro et al., 2016), "*gplots*" for creating plots (Gregory et al., 2009), and "*gdata*" for importing input data from MS Excel formatted files (Gregory et al., 2017). Unless indicated otherwise, the significance level was set to $\alpha = 0.05$.

## 3. Results

### 3.1 Meteorological and terrain conditions

**Kraichgau**

At the KR sites, the mean air temperature ranged between 8.4 °C in 2010 and 10.9 °C in 2014. On average it was 9.8 °C (Fig. 2a). This value is 0.4 °C higher than the 30-y climatological mean (1981−2010) measured at the meteorological station Pforzheim-Inspringen. The lowest and highest monthly mean temperature was −3.2 °C in February 2012 and 21.1 °C in July 2015. The mean annual precipitation was 796 mm, which is 93 mm lower than measured in Pforzheim-Inspringen. In 2013, the wettest year within the 8-y period, total precipitation amounted to 973 mm. The lowest annual precipitation (629 mm) was measured in 2015 (Fig. 2b).

Typical for the mid latitudes, the KR sites' prevailing wind direction was from west to east. The fraction of WSW to WNW (240° – 300°) winds was 43.2, 36.8 and 33.7 % at EC1, EC2 and EC3, respectively (Fig. 3). The highest wind speeds were also measured within these wind direction sectors. Wind blowing from north- and southward directions was rarely measured (< 10 %). While at EC1 the wind speed averaged 2.9 m s$^{-1}$, at EC2 and EC3 the values were only 2.4 and 1.9 m s$^{-1}$, respectively. Moreover, high wind speeds (> 6 m s$^{-1}$) clearly decreased from west to east. At EC1, EC2 and EC3, the share of these high wind speeds was 4.6, 3.3 and 0.2 %, respectively.

Fig. 4 shows the height transects along the prevailing wind direction. At the KR sites, the mean slopes along the transects were 0.4 %, <0.01 % and 0.3 % at EC1, EC2 and EC3, respectively. The micro-relief of station EC1, located on a micro-bank, fluctuates more strongly than that of EC2. The immediate surroundings of EC2 are very homogeneous in elevation. Station EC3 is positioned in a micro-depression. Overall, the three transects show that the KR fields can be regarded as nearly flat, with EC2 being the flattest.

**Swabian Jura**

The mean temperature in SJ (7.4 °C) was 2.4 °C lower than in KR. It varied from 5.9 °C in 2010 to 8.5 °C in 2015 (Fig. 2c). As in KR, the lowest and highest mean monthly temperatures were recorded in February 2012 (−6.6 °C) and July 2015 (18.6 °C). The mean annual precipitation was 874 mm. As in KR, 2015 was the year with the lowest precipitation. Highest total





rainfall was measured in 2017, not in 2013 as in KR. November 2011 was the month with the lowest total precipitation (5 mm), and July 2014 that with the highest (187 mm) (Fig. 2d).

In SJ, the wind blew mostly from westerly or easterly directions (Fig. 3). The wind from the 240°−300° sector was less than in KR, with shares of 14.4, 25.5 and 26.6 % at EC4, EC5 and EC6, respectively. At EC5, more wind was recorded from the

NW sector (300–330°). Mean horizontal wind speeds at EC4, EC5 and EC6 were 2.44, 2.38 and 2.51 m s$^{-1}$, respectively. Wind speeds above 6 m s$^{-1}$ made up 2.0 (EC4), 1.7 (EC5) and 1.3 % (EC6) of all measured wind speeds in SJ.

In SJ, only EC4 is relatively flat (Fig. 4). Its topography is comparable with that of EC1 in KR. The elevation along the transect at EC5 gently increases, with a mean slope of 0.6 % from SE to NW. Station EC5 itself is situated in a local micro-depression. The topography of station EC6 differs considerably from that of the other fields. The station is positioned on the

top of a ridge. Whereas in NW direction the terrain drops with a mean rate of 3.7 m per 100 m, in SE direction the terrain is nearly flat (slope = 0.3 %).

## 3.2 Energy partitioning at the land surface

The energy partitioning at the canopy surface of different crop stands is shown, by way of example, for the vegetation period

of 2016 (Fig. 5). In that year, five different crops (winter rapeseed (WR), spring barley (SB), winter wheat (WW), silage (SM) and grain maize (GM)) were grown at the EC sites. From April to June, most of the net radiation was transformed into latent heat at the crop stands, except for the silage maize at EC5. The daytime Bowen ratio was lowest for WW and WR, with 0.14 and 0.13, respectively. Also GM, SB and SM at EC6 led to daytime Bowen ratios distinctly below unity (about 0.21). Only silage maize at EC5 had a Bowen ratio of about unity, which indicates that the available energy was partitioned

into latent and sensible heat in similar proportions. At the cereal and WR sites and years, the ground and sensible heat fluxes were nearly the same and showed a similar diurnal course. At the maize stands, the ground heat flux tended to be higher than the sensible heat flux during the morning hours, while in the afternoon the order switched and more sensible heat than ground heat was formed. At all sites, the energy residual was similar to the sensible heat flux, ranging from 23 W m$^{-2}$ at EC3 to 44 W m$^{-2}$ at EC1. The daily net radiation was 149, 133, 134, 130, 138, and 164 W m$^{-2}$ at EC1−6, respectively. The mean

daily *LE* ranged from 54 W m$^{-2}$ at EC5 to 94 W m$^{-2}$ at EC3.

For July to September, the strongest shift in energy partitioning occurred at the WR site. Over noon, the Bowen ratio was in the range of unity, and sometimes the half-hourly sensible heat flux was even higher than the latent heat flux. A similar shift was observed at the WW site, but weaker than at the WR site. At the GM, SM and SB sites the largest difference compared with the period April to June was the ratio between sensible and ground heat flux. From July to September the sensible heat

was about twice the ground heat flux. The mean net radiation ranged from 125 W m$^{-2}$ at EC5 to 176 W m$^{-2}$ at EC2, and *LE* varied from 78 W m$^{-2}$ at EC4 to 89 W m$^{-2}$ at EC1. The residual energy for this period was 23, 28, 22, 24 and 16 W m$^{-2}$ at the sites EC1, EC2, EC3, EC4, and EC6, respectively. Note: EC5 data are missing due to sonic anemometer and gas analyser damage.



### 3.3 Energy balance closure

The mean EBR over the 48 site-years was 0.75, corresponding to a mean energy residual of 41.6 W m$^{-2}$ (Tab. 3). The mean annual EBR ranged between 0.62 at EC1 (WW in 2013) and 0.90 at EC4 (SM in 2017). The mean EBR over the six EC stations was highest in 2010 (EBR = 0.78) and lowest in 2013 (EBR = 0.71). Averaged over the period from 2010 to 2017, the best EBC was achieved at EC4 (EBR = 0.82), whereas the largest mean energy gap occurred at the neighboring station EC5. There, the mean residual was 49.0 W m$^{-2}$.

Figure 6 presents the course of monthly mean EBCs determined by *OLR* for all six stations averaged over the period 2010−2017. In general, the EC method performed best (EBC was highest) over the vegetation period from April to August. The highest EBC was usually obtained in July and August. The EBC distinctly declined over autumn and winter. At station EC6, the SJ station equipped with a fuel cell system, for example, the EBC declined to 42 % in January 2016 and 23 % in December 2017. A low EBC was usually associated with a larger variation (see winter months; Fig. 6).

### 3.4. What impacts the EBC?

### 3.4.1 Effect of region, station, year and crop

The statistical analyses showed that the EBC did not differ between the two regions (Fig. 7a) over the main vegetation period from April to June. The EBC measured at stations EC2 and EC4 was significantly higher ($p < 0.001$) than at the other stations (Fig. 7b). The lowest spread in values was observed at station EC4. In 2013 and 2014, EBC was lower ($p < 0.001$) than in the other six years (Fig. 7c). The crops had no significant effect on mean EBC (Fig. 7d). EBC over winter rapeseed showed the highest variation among the four crops, varying between 57 and 88 %.

### 3.4.2 Effect of wind direction

The distribution of EBR as a function of wind direction is shown in Fig. 8. The EBR was averaged for 30° wind sectors over all available daytime data (global radiation >10 W m$^{-2}$). At the KR sites, the highest EBR was achieved when the wind blew from westerly directions, which is the prevailing wind direction. Wind from northern and southern directions was related to lower EBR. Particularly wind from south was associated with EBR below 0.6 at all three stations. This phenomenon was most pronounced at station EC1. Also at the SJ sites, highest EBR was usually linked to the prevailing wind direction. One exception is the high EBR at EC4 for the SSE sector. At EC4, for six out of twelve wind sectors, EBR was above 0.8. At




EC5 and EC6, in contrast, the EBR exceeded 0.8 in only three wind sectors. At all stations, EBR was lowest (< 0.6) for winds from north-east (Fig. 9).

### 3.4.3 Effect of atmospheric conditions

This section evaluates the EBR as a function of buoyancy, shear and atmospheric stability. For this purpose, we plotted EBR against kinematic virtual temperature flux ($w'T_v'$, proxy for buoyancy), friction velocity ($u^*$, proxy for shear) and the stability parameter $\zeta$ (Fig. 10). Again, only half-hourly daytime fluxes ($R_s > 10$ W m$^{-2}$) were evaluated. The plot EBR versus $w'T_v'$ shows a vast scatter at weak buoyancy, i.e. during the transition periods between positive and negative $w'T_v'$. Here, the EBR ranges from plus four to minus four. The scatter decreases substantially as the modulus of $w'T_v'$ increases. Note that

$w'T_v' < -0.15$ K m s$^{-1}$ were measured only at stations EC2 and EC4.

Plotting EBR against friction velocity also reveals a large scatter, which narrow as friction velocity (shear) increases. The scatter, however, does not narrow as much as for increasing buoyancy. During neutral or stable atmospheric conditions, the EBR showed a large spread (Fig. 10). In contrast, this range distinctly declined when the stability parameter reached strongly negative values, indicative for highly unstable conditions. EBR above unity or below zero was rarely observed under these

conditions.

From the total dataset, only 7 % of daytime measurements were made under stable conditions, 34 % under unstable conditions and 59 % under neutral conditions (Table 4). During unstable conditions, EBC and EBR at all sites were slightly lower compared to neutral conditions. During neutral conditions, however, the standard deviation (SD) of EBR was about twice as high as under unstable conditions. During stable conditions, EBC and EBR were systematically lower than at

unstable and neutral conditions. At EC4, for example, the EBR under neutral and unstable conditions was 0.78 and 0.85, respectively. Under stable conditions, the value declined to 0.41. Moreover, the huge spread in the EBR under stable conditions is underlined by its high SD, which is about three times the mean value.

### 3.4.4 Effect of footprint

Fig. 11 shows exemplary footprints for sites EC3 and EC5 in 2015, illustrating the substantially different situations observed in different months. Both fields were cropped with maize. At EC3, EBC continuously increased from 68 % in June to 79 % in July and 90 % in August. In this period, as the maize plants got taller, the footprint area became continuously smaller. A similar relation between footprint area and EBC was observed at EC5: the larger the footprint the lower EBC. A linear regression between EBC and the 90 % footprint area of all data from 2015 confirmed this relation (figure not shown).

Although $R^2$ was only 0.21, the slope of $-1.25$ % ha$^{-1}$ (0.50 % ha$^{-1}$; standard error) per hectare was significantly different from zero. The intercept of the regression was 79 %.



## 4. Discussion

### 4.1. EBC and energy balance components

From July to September, daily mean $R_n$ varied between 125 and 176 W m$^{-2}$. Similar ranges of $R_n$ were observed with maize
in Livraga, Italy (Masseroni et al., 2014). The latent and sensible heat fluxes varied strongly over the observational period. In
early-covering crops (winter rapeseed, winter wheat, winter barley), $LE$ was about two to three times higher than $H$ in the
period AMJ (April-May-June), while in the period JAS (July-August-September) $LE$ and $H$ were in a similar range (Fig. 5,
EC3-WW, EC4-SB). The period JAS represents ripening and harvest of cereals and winter rapeseed as well as post-harvest
management such as tillage and seeding of cover crops or winter rapeseed. During AMJ, the patterns of $LE$ and $H$ at EC5
and EC6 differed, even though the maize was sown on similar days of the year (May 07 at EC5 and May 03 at EC6). This
can be explained by the substantially higher leaf area index at EC6 (0.74±0.15) compared to EC5 (0.35 ±0.06), measured on
June 22.

The mean EBR of the 48 site-years was 0.75 (Table 3). In comparison, Wilson et al. (2002) reported an EBR of 0.84, on
average, for the 50 analyzed FLUXNET site-years, ranging from 0.34 to 1.69. In three agricultural and one industrial site in
South Korea, the mean value varied between 0.46 and 0.83 (Kim et al., 2014). Majozi et al. (2017) found a mean of 0.93 at a
semi-arid savannah site in South Africa, over a period of 15 years.

The slopes of the OLR and EBR differed by a maximum of 5 %, which is consistent with previously published data. Such a
small difference between both measures points at a high reliability of the presented EC measurements (Wilson et al., 2002).
The highest annual EBC occurred at EC4 (87 % in 2010), the second highest at EC2 (83 % in 2016), the lowest at EC5 (62
% in 2016). The lowest EBC was observed mainly in the cold, non-growing season, which may be attributed to insufficient
thermally and mechanically induced turbulence (Franssen et al., 2010) as well as to freezing (Varmaghani et al., 2016).

The incomplete EBC in our dataset has several potential explanations. One is the neglected minor storage terms (Eshonkulov
et al., 2018; Masseroni et al., 2014; Meyers and Hollinger, 2004). Importantly, considering minor storage terms is not
straightforward because they are not measured when conventional EC equipment is used. Only the energy consumed and
released by photosynthesis and respiration can be directly derived from EC data because the net $CO_2$ flux is generally
measured. Considering minor storage terms in calculating the EBC at a maize field improved the mean value from 87 to 91
% (Xu et al., 2017) and from 81 to 86 % (Masseroni et al., 2014). Eshonkulov et al. (2018) demonstrated that minor storage
and flux terms over winter wheat in southwest Germany contributed the most to the EBC during the main vegetation period
in May. During this month the minor terms helped to close the energy balance by an additional 7−8 %.



## 4.2 The effect of meteorological conditions and surface-layer turbulent parameters

In both KR and SJ, EBR was highest for winds blowing from the prevailing wind direction. This is consistent with other studies. Xin et al. (2018), for example, found that winds blowing from the prevailing direction yielded consistently higher EBC compared to other directions. Kim et al. (2014), for example, grouped EBR into two different categories, with lower (<

0.75) and higher EBR (> 0.75). At their four research sites, the authors observed that EBR was higher at high wind speed.

At the SJ sites, we found particularly low EBRs in the wind sector 0−90°. The CSAT sensor was oriented mostly to 225°, so that the sector 30−90° was located behind the anemometer head. To substantiate the idea that the anemometer negatively influences EC measurement quality, and taking the data from EC4 as an example, we recalculated EBR across all years, excluding the wind directions of the sector 0−90°. This increased the mean EBC in 2010−2017 by 4 percentage points from

80 to 84 % (data not shown). Friebel et al. (2009) used a wind tunnel experiment to show that there is a 40° shadow zone behind the sonic anemometer where the measured wind speeds were reduced by up to 16 %. Within a shadow zone of about 20° behind the anemometer, the turbulent spectra were corrupted. Our findings indicate that under field conditions the shadow zone is even somewhat wider (about 60°). A practical solution for measuring reliable fluxes when winds blow from the back of the anemometer could be to operate an anemometer tandem: a first anemometer orientated in the prevailing wind

direction, a second one in the opposite direction. Whether this set-up could solve the problem requires further investigation.

Fig. 10 shows that the spread of EBR distinctly narrowed at high friction velocities ($u^* \geq 0.5$). Prior studies have noted the importance of $u^*$ on the EBC. Anderson and Wang (2014) found that, under these conditions, EBC was closed on days with continuous turbulence. Their results confirm that their EC site had various turbulence and closure patterns. Results of hourly daytime EBR and $u^*$ showed a strong relationship at our sites (Fig. 10). This is consistent with other studies carried out in

selected croplands such as irrigated sugarcane (Anderson and Wang, 2014), maize plantations (Masseroni et al., 2014) and rice fields (Kim et al., 2014). Sánchez et al. (2010) also reported that EBR was  > 0.90 when high friction velocities prevailed (> 0.8 m s$^{-1}$) at a boreal forest site in Finland. Mauder et al. (2013) investigated EBC at the TERENO site Lackenberg (Germany) and found that it was almost closed. They explain this result by the very good turbulent mixing and the high homogeneity at this site. This confirms that, at high $u^*$, the production of high-frequency fluxes is elevated (Fratini

and Mauder, 2014).

At our study sites, neutral conditions dominated at ~60 %, followed by ~34 % unstable and 6 % stable conditions (Table 4). Importantly, average EBR changed from 0.67 (± 0.32) to 0.72 (± 0.69) and 0.41 (± 1.33) during unstable, neutral, and stable conditions, respectively (SD in brackets). Under stable conditions, the EBR was lowest and had the largest variation. Averaged over all EC stations, the slope of OLR under neutral conditions was slightly higher than under unstable conditions.

This is also evident in the calculated energy residual and its SD. The average residuals under stable, neutral and unstable conditions were 9.7 (± 31.5), 47.5 (± 58.2) and 81.5 (± 62.0) W m$^{-2}$, respectively. The coefficient of variation was highest under stable conditions and decreased over neutral to unstable conditions. This result differs from previous studies. Mauder et al. (2010) reported a residual energy close to zero under stable conditons, peaking at 150 W m$^{-2}$ under neutral, and





decreasing to 100 W m$^{-2}$ under unstable conditions in cropland in Ontario, Canada. Note, however, that the difference of residual energy under stable conditions may be the result of using only daytime data (from 7 am to 7 pm).

The scatter of EBR versus buoyancy flux at EC2 and EC4, the two stations with the highest EBC, differed from those of the other stations (Fig. 10). At these two sites, strong negative buoyancy fluxes below −0.15 K m s$^{-1}$ were recorded. This means

that the atmosphere was not heated by the land surface, but that the land surface was significantly heated by the atmosphere. Our finding of the highest EBR at the two sites with the most pronounced buoyancy does not fit well with studies that recommended considering secondary circulations to achieve a better EBC (Cava et al., 2008; Foken et al., 2006; Kidston et al., 2010, Mauder et al. 2010). Those studies postulate that heterogeneity-induced and buoyancy-driven quasi-stationary circulations are probably the dominant processes behind underestimated energy fluxes. In considering secondary

circulations, different time averaging intervals instead of the standard 30-min period can be used. Although a 60-min interval might be suitable for capturing the major turbulent fluxes (Kilinc et al., 2012), in most cases the standard 30-minute period is still sufficient (Kidston et al., 2010). Finding an optimum averaging period is a very complex to nearly impossible task. This is because atmospheric turbulence changes irregularly and there is no clear-cut "switch" in time. Therefore, the averaging time could be modified during raw data processing. In practice, however, this is unlikely because it drastically increases the

complexity of data processing (Lenschow et al., 1994). Moreover, the sources of secondary circulations are unclear, and they are most probably not well linked with the locally measured available energy. Accordingly, excluding secondary circulations in EC measurements can be locally meaningful. Recently, a new method, known as ogive optimization, was proposed by Sievers et al. (2015). The method enables separating low-frequency influences from vertical turbulent fluxes to isolate the local exchange processes of interest.

Although EC measurements contain uncaptured energy components, the flux data are used, among others, to evaluate models and interpret simulation results. In such studies, EC flux data are usually post-closed, i.e., the measured turbulent fluxes are adjusted so as to close the energy balance (Ingwersen et al., 2015). The standard approach is the Bowen-ratio post-closure method (Twine et al., 2000). It assumes that the missing energy has the same Bowen ratio as the measured turbulent fluxes. This approach, however, may introduce a systematic bias to simulated surface energy fluxes (Chen and Li, 2012).

Analyses of the energy imbalance by Ingwersen et al. (2011) showed that soil water contents simulated by a land surface model agreed better with  measurements when the residual was fully assigned to $H$. As discussed by Charuchittipan et al. (2014), secondary near-surface circulations attributed to low frequencies mainly transport sensible heat. Therefore, they proposed a new alternative energy balance correction method they termed the Buoyancy flux ratio. At very large Bowen ratios (> 10), the Bowen ratio post-closure and buoyancy flux correction methods yield similar results. At Bowen ratios

ranging from 0.1 to 0.2, which is typical for croplands during the main growing period, the Buoyancy flux ratio method assigns most of the energy residual (> 50 %) to the sensible heat flux. The Bowen ratio method, in contrast, distributes most of it (> 90 %) to latent heat. As long as the composition of the residual remains unknown, it is important to communicate the possible error in EC flux data, for example with the post-closure method uncertainty band (PUB) (Ingwersen et al., 2015). Working with only one post-closure method may result in serious misinterpretations in model–data comparisons.





### 4.3 Relationships between EBC and footprint

Accurate measurements of energy balance components are important to achieve a good EBC. In this context, one key requirement is that the EC station be located in a place that represents the fluxes from the area of interest (Burba and

Anderson, 2010). According to those authors, the terrain must be horizontal and uniform. Three parameters are needed in footprint analysis: measurement height, surface roughness and atmospheric stability. When turbulent fluxes originate from a horizontal and homogeneous surface, the footprint depends solely on the distance between the location of the measurement point and the emission element. We found a distinct tendency that the smaller the footprint, the higher the EBC. We forward two explanations. First, the smaller the footprint, the higher the chance that the assumption of a homogeneous source area is

fulfilled. Second, the smaller the footprint, the better the scale match between the measurement of available energy and turbulent fluxes. Alfieri and Blanken (2012) found that variations of surface energy fluxes over tens of meters ranged from 30 to 40 W m$^{-2}$ using single-point (immobile) and mobile EC towers at a uniform site (Colorado, USA). They concluded that a single-point EC tower cannot capture all the relevant energy fluxes because they vary spatially. Our results confirm that if the footprint is small, the EBC from EC measurements is better, which can be interpreted as a reduction in the variation of

surface energy fluxes.

Many studies claimed that surface heterogeneity is a potential reason for the energy imbalance (Stoy et al., 2013; Xu et al., 2017). The latter authors reported that EBC decreased with increasing surface heterogeneity. The degree of heterogeneity was derived from high-resolution remote sensing images and land surface temperatures. To handle this effect, some authors recommend using direction-specific coefficients that indicate the degree of heterogeneity. For example, Panin et al. (1998)

introduced a heterogeneity factor that comprises surface parameters such as roughness, radiation and the thermal humidity of the internal boundary layer. That factor can be used for data interpretation. Nonetheless, deploying heterogeneity factors still does not explain how the residual energy is composed. The lack of EBC at the KR sites might partly reflect katabatic advection (Kutsch et al., 2008), which results from stable atmospheric conditions and occurs especially in hillslope areas (Loescher et al., 2006; Mauder et al., 2010). The former waste dump located about 500 m south of the fields in KR (Fig.1)

might be responsible for advective fluxes. It is approximately 41 m higher than the study sites. Moreover, the topography could also affect EBC. The elevation transects show that the immediate terrain surrounding the stations EC3, EC5 and EC6 is not totally flat (Fig. 4). This is a well-known problem for micro-meteorological field measurements (Wilczak et al., 2001). At EC1, EC2, and EC4, however, the terrain can be considered flat.



## Conclusions

We evaluated the EBC of long-term EC measurements at six different cropland sites in two contrasting environmental regions in southwest Germany. EBC depended on how well thermally and mechanically induced turbulence was developed. On average, 25 % of the available energy was not detected by our EC stations, with the lowest annual imbalance (energy residual) of 17 % in KR and 13 % in SJ. This range of EBC is common in cropland, and such recovery rates must be accepted in heterogeneous landscapes. We interpret the range of the highest mean annual EBC (83 % resp. 87 %) as the upper detection limit of the EC method at our sites and settings. During winter months and under stable atmospheric condition, EBC was problematic. EBC was negatively affected by: i) stable atmospheric conditions, ii) non-horizontal or heterogeneous source area, iii) larger obstacles in the landscape, i.e. the former waste dump that may have induced adjective flux components, and iv) flow distortions of winds first traveled past the back head of the anemometer, which reduces wind speed and corrupts the spectral characteristics of turbulence at specific wind directions. EBC was positively affected as the footprint area decreased, probably because this tends to decrease the heterogeneity of the source area and improves the match of available energy measured locally with the mean available energy in the footprint.

## Acknowledgements

This study was financially supported by the German Research Foundation (DFG) in the frame of the Research Unit (RU) 1695 "Structure and function of agricultural landscapes under global climate change – Processes and projections on a regional scale". Part of the work was sponsored by an Erasmus Mundus grant "TIMUR − Training of Individuals through Mobility from Uzbek Republic to EU (referenced as GA NO 213−2723/001−001−EM Action 2)". We would like to thank the farmers Mr. Bosch senior (†), Mr. Bosch junior (EC1, EC2 and EC3) in KR, Mr. Fink (EC4), Mr. Hermann (EC5) and Mr. Reichart (EC6) in SJ for the permission to conduct measurements in their fields. The authors would also like to thank the technical staff Benedikt Prechter, Felix Baur, Christian Schade, and Thomas Schreiber.

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

**Figure 7.** Comparison of energy balance closure (EBC) measured by linear regression grouped for the different regions, sites, years and crops. Measurements were conducted from early spring until harvest. Different letters indicate significant differences between the factor levels at α = 0.05.

**Fig. 8.** Distribution of half-hourly energy balance ratios (EBR) in terms of wind direction at eddy covariance (EC) stations during the 2010 to 2017study period. Spike lengths in diagram show relative frequency of wind directions; color of legend shows EBR. Upper panel: Kraichgau: Lower panel: Swabian Jura.

**Fig. 9** Half-hourly energy balance ratio (EBR) averaged for 30° wind sectors at the six eddy covariance (EC) stations during the 2010 to 2017 study period. Upper panel: Kraichgau: Lower panel: Swabian Jura.

**Figure 10.** The mean energy balance ratio (EBR) as a function of buoyancy flux ($w'T_s'$), friction velocity ($u*$) and the stability parameter (ζ) during the 2010 to 2017study period.

**Figure 11.** Footprint area of EC3 and EC5 in selected months of 2015 and the corresponding energy balance closure (EBC). Black points represent positions of EC stations. Yellow lines indicate relative areal contributions to total flux in 10 % steps, where the outmost yellow line indicates the area from which 90 % of measured fluxes originated. The satellite image was taken from Google Earth (images from 31.03.2017 and 3.30.2014 for EC3 and EC5, respectively).

**List of Tables**



**Figure 1. a) Geographical overview (a) and locations of the study sites and EC stations in Kraichgau (b) and Swabian Jura (c). Satellite images were taken from Google Earth (of KR from 31.03.2017 and of SJ from 26.08.2016). Red transect lines indicate positions of conducted micro-topographic measurements along the prevailing wind directions. Area bordered with yellow line at Kraichgau sites in (b) shows former waste dump.**





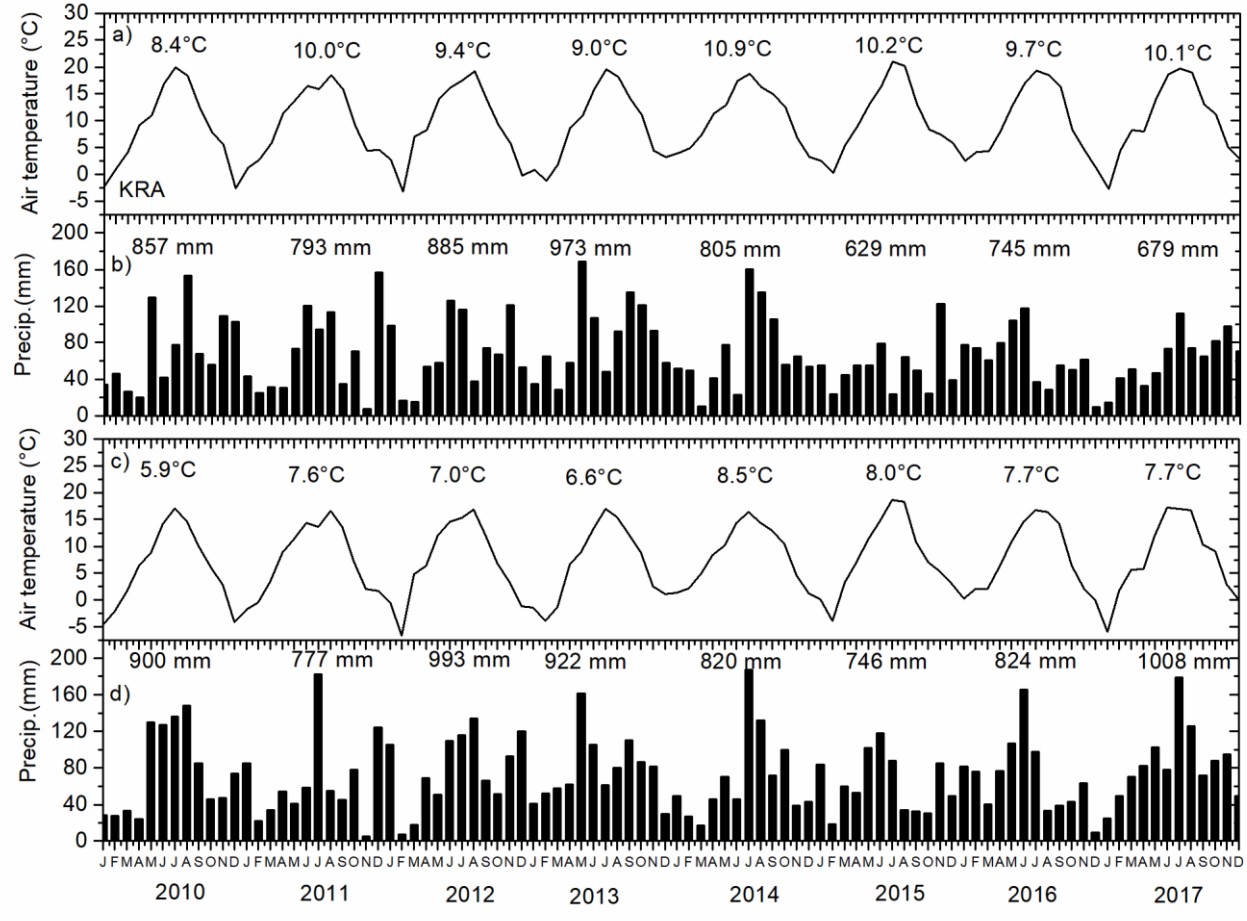

**Figure 2. Mean monthly air temperatures and precipitation sums at the Kraichgau site EC1 and Swabian Jura site EC4 from 2010 to 2017. Annual mean temperatures and precipitation sums are given on top of the lines/bars.**





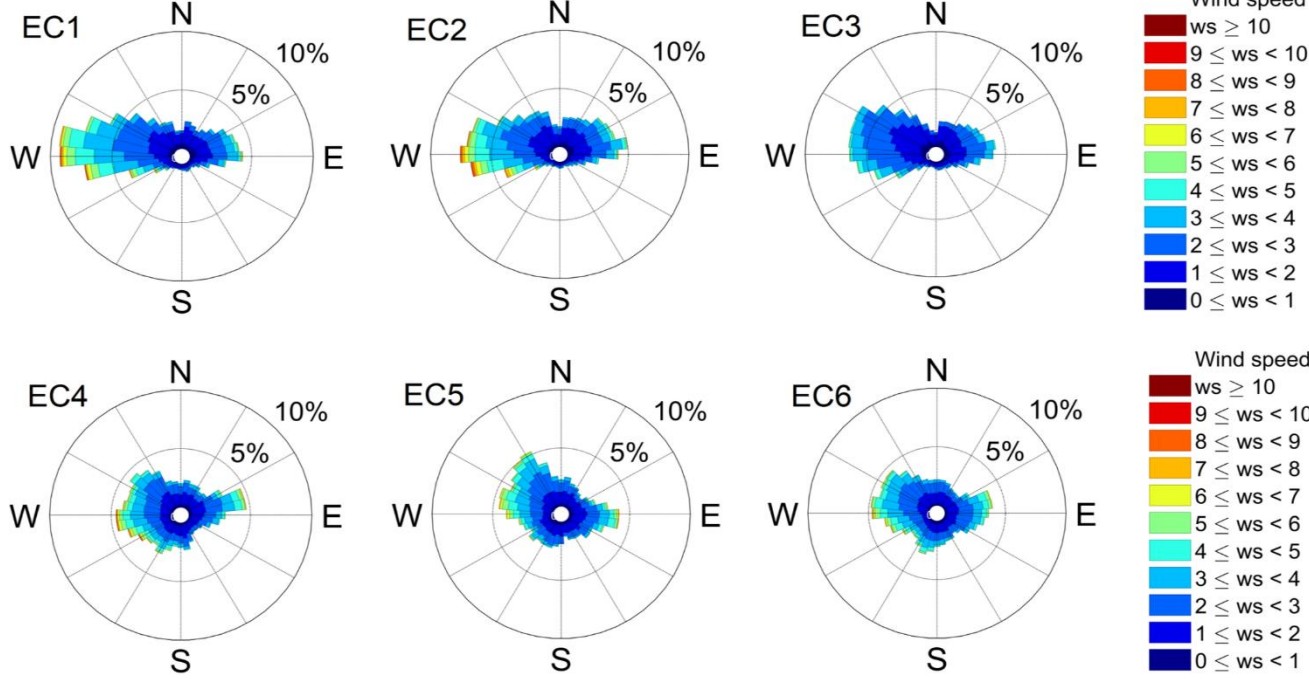

**Figure 3. Distribution of wind direction and wind speed (m s$^{-1}$) from 2010 to 2017 in Kraichgau (EC1 – EC3) and Swabian Jura (EC4 – EC6).**





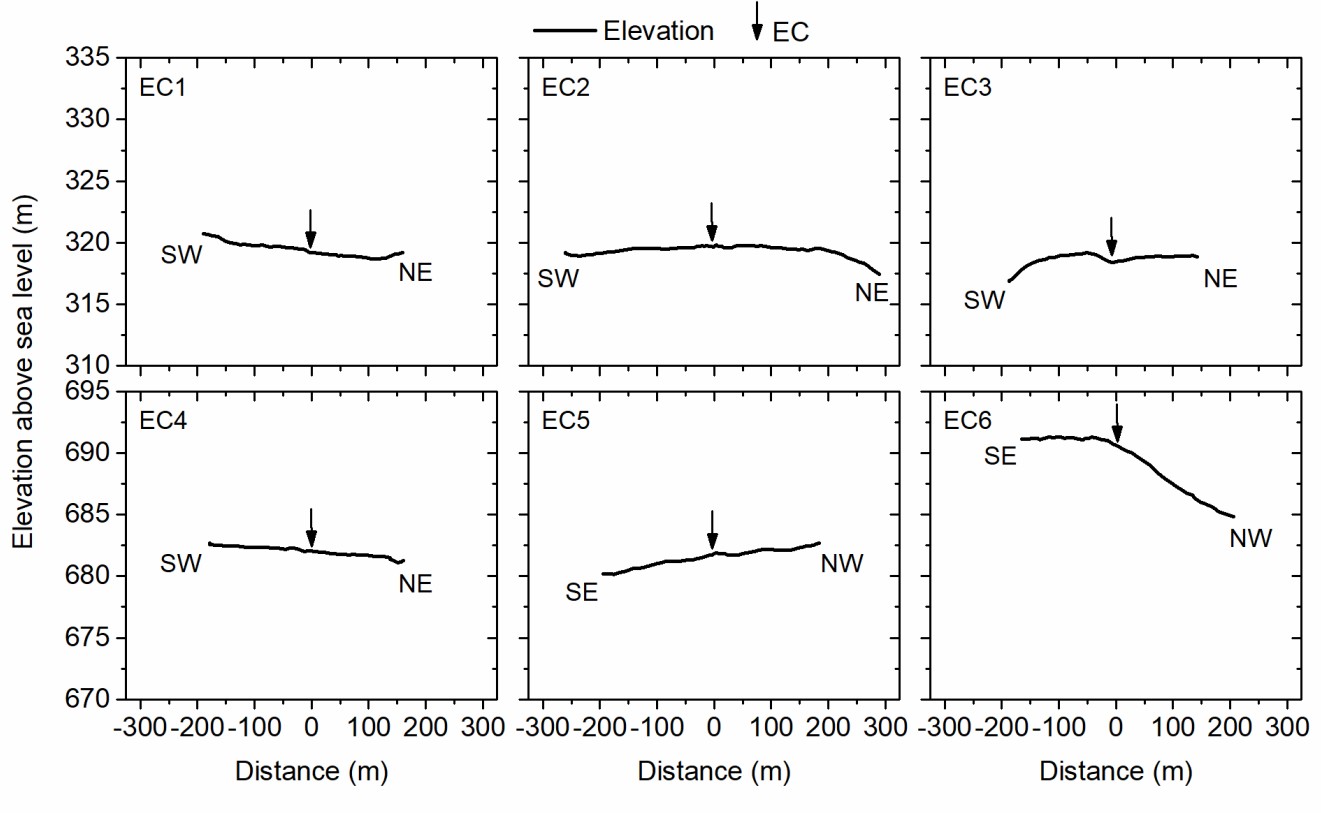

**Figure 4. Elevations at EC sites along the dominant wind directions (see Figs. 1 and 3). Arrows present positions of EC stations.**





**Figure 5. Diurnal courses of energy balance components averaged over 3-month periods in Kraichgau (EC1, EC2, EC3) and Swabian Jura (EC4, EC5, EC6) in 2016. Insets denote the different crops grown in 2016; for explanation see main text. Because of energy shortage during winter at the EC1, EC3, EC4 and EC5 sites, the fluxes shown in the JFM and OND graphs were measured only in March and from 1 October to mid-November, respectively.**



**Figure 6. Monthly aggregated energy balance closure (EBC) obtained by ordinary linear regression of turbulent fluxes ($LE+H$) against available energy ($R_n-G$) for all stations during 2010−2017.**



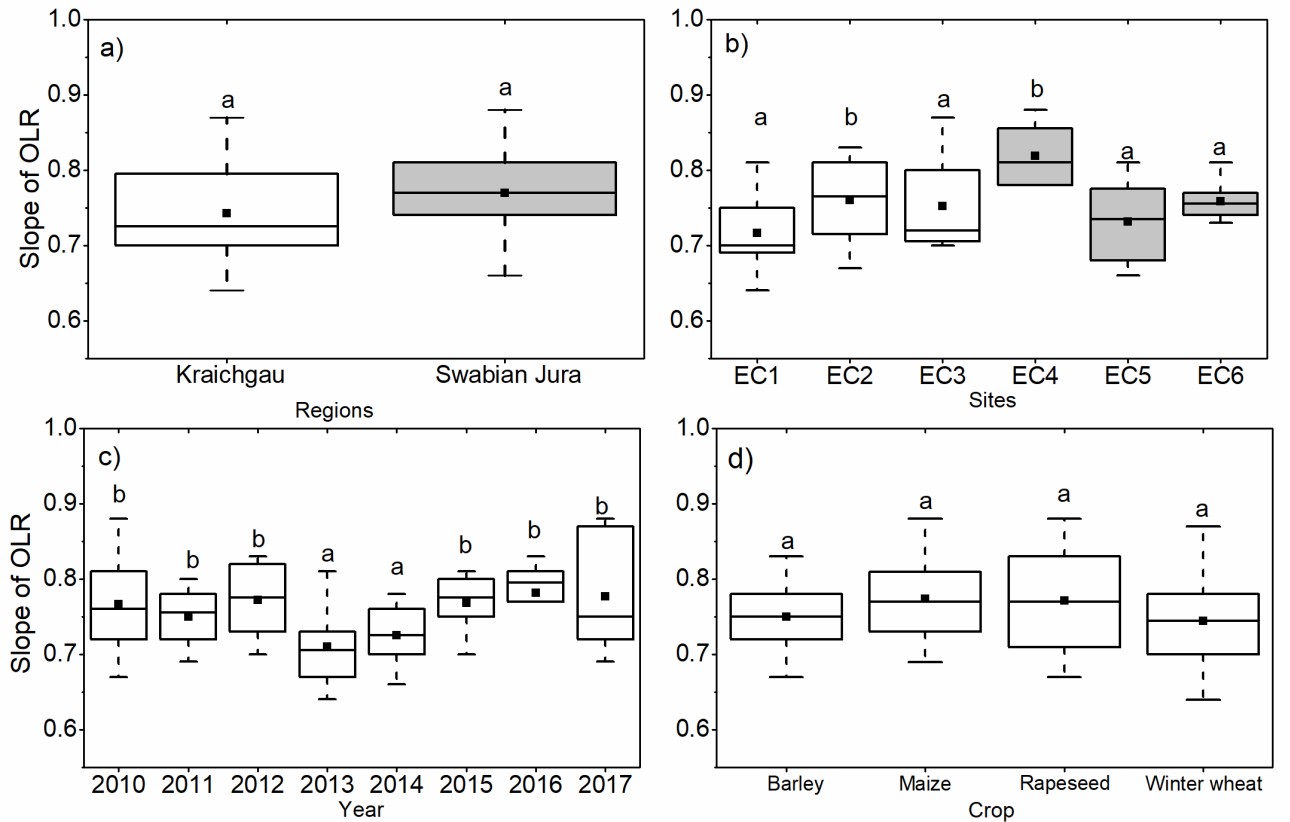

**Figure 7.** Comparison of energy balance closure (EBC) measured by linear regression grouped for the different regions, sites, years and crops. Measurements were conducted from early spring until harvest. Different letters indicate significant differences between the factor levels at $\alpha = 0.05$.




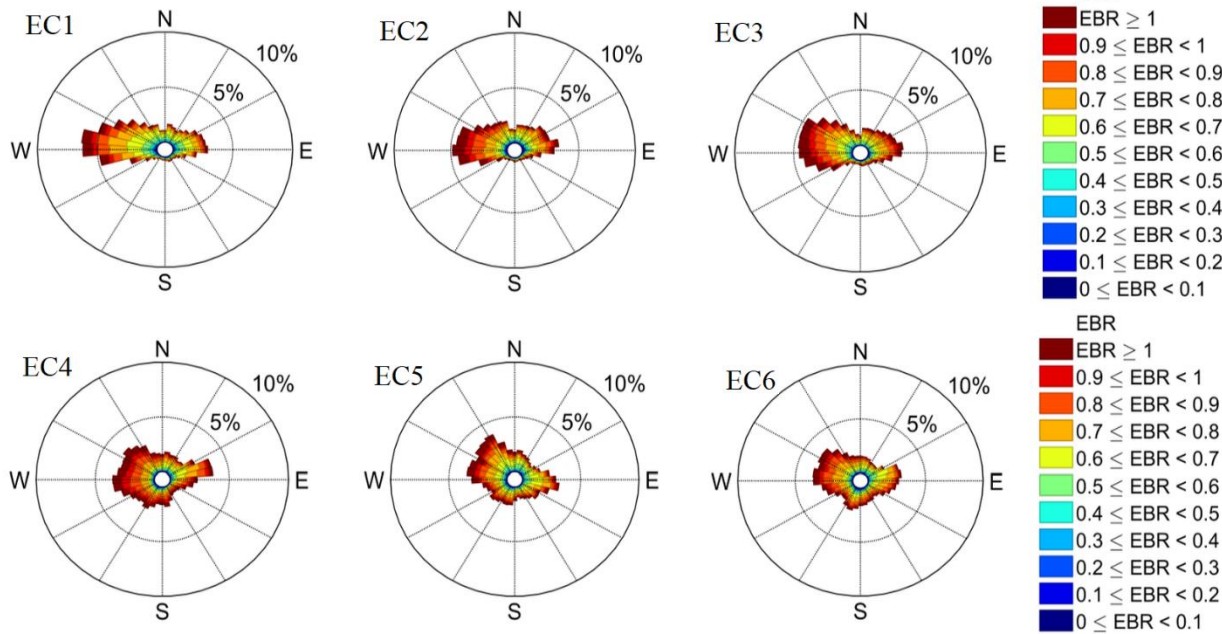

**Figure 8. Distribution of half-hourly energy balance ratios (EBR) in terms of wind direction at eddy covariance (EC) stations during the 2010 to 2017 study period. Spike lengths in diagram show relative frequency of wind directions; color of legend shows EBR. Upper panel: Kraichgau: Lower panel: Swabian Jura.**





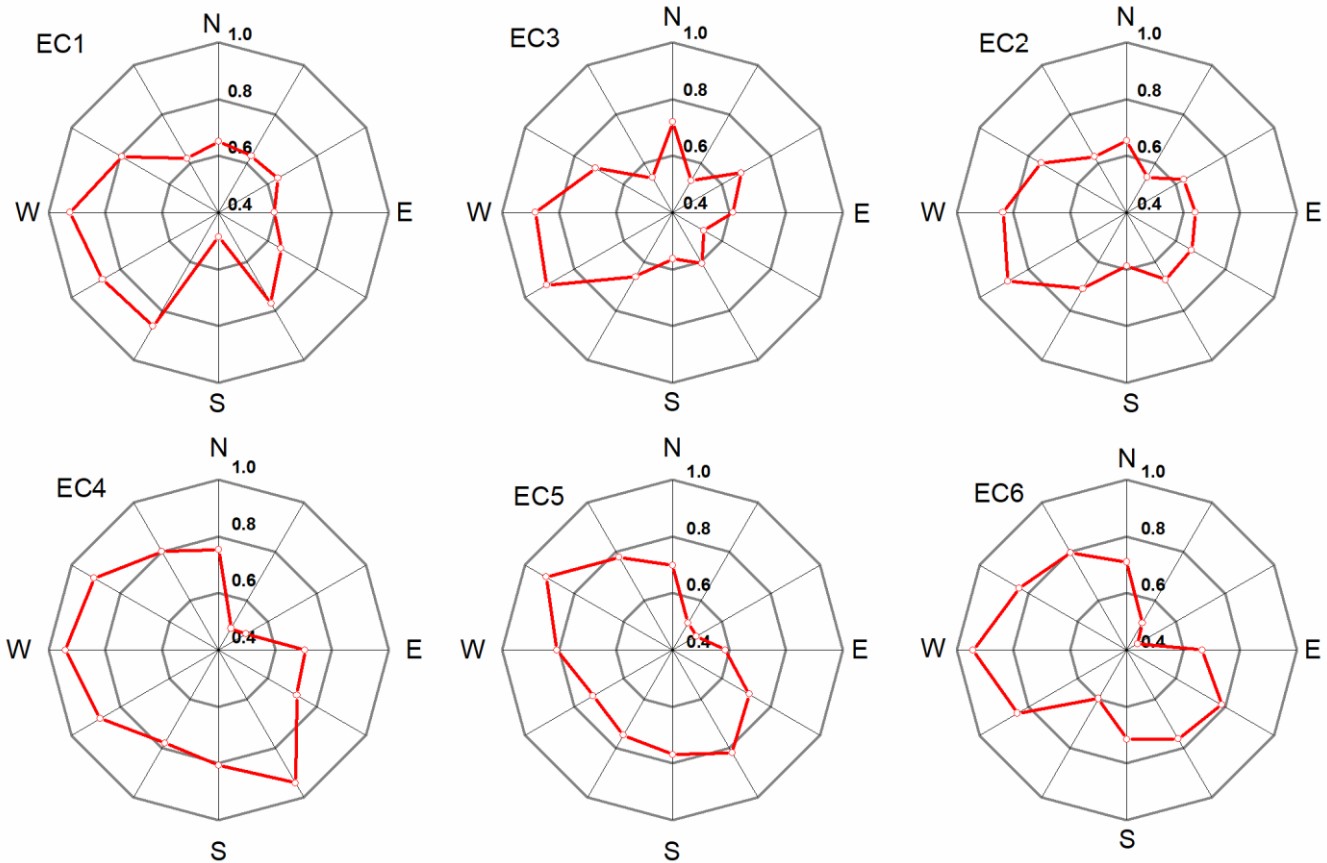

**Figure 9. Half-hourly energy balance ratio (EBR) averaged for 30° wind sectors at the six eddy covariance (EC) stations during the 2010 to 2017 study period. Upper panel: Kraichgau: Lower panel: Swabian Jura.**





**Figure 10. The mean energy balance ratio (EBR) as a function of buoyancy flux ($w'T_s'$), friction velocity ($u^*$) and the stability parameter ($\zeta$) during the 2010 to 2017 study period.**





**Figure 11. Footprint area of EC3 and EC5 in selected months of 2015 and the corresponding energy balance closure (EBC). Black points represent positions of EC stations. Yellow lines indicate relative areal contributions to total flux in 10 % steps, where the outmost yellow line indicates the area from which 90 % of measured fluxes originated. The satellite image was taken from Google Earth (images from 31.03.2017 and 3.30.2014 for EC3 and EC5, respectively).**





**Tables**

**Table 1.** Main characteristics of the investigated sites.

| Region | Kraichgau | | | Swabian Jura | | |
|---|---|---|---|---|---|---|
| Station name | EC1 | EC2 | EC3 | EC4 | EC5 | EC6 |
| Latitude (°) | 48.928496N | 48.927743N | 48.927199N | 48.527214N | 48.529780N | 48.546632N |
| Longitude (°) | 8.702782E | 8.708901E | 8.715950E | 9.769950E | 9.773474E | 9.774280E |
| Elevation (m) | 319 | 320 | 319 | 682 | 681 | 690 |
| Soil type, (WRB, 2014) | Stagnic Luvisol | | | Calcic Luvisol | Anthrosol | Rendzic Leptosol |



**Table 2.** Crop grown at the four study sites from 2010 to 2017 (harvest year).

| Region | Kraichgau | | | Swabian Jura | | |
|---|---|---|---|---|---|---|
| | Sites | | | Sites | | |
| Harvest year | EC1 | EC2 | EC3 | EC4 | EC5 | EC6 |
| 2010 | SM | WR | WW | WR | WW | SM |
| 2011 | WW | WW | SM | WW | SM | WW |
| 2012 | WR | SM | WW | SB | SM | WB |
| 2013 | WW | WW | WR | WR | WB | SM |
| 2014 | SM | SM | WW | WW | SP | WW |
| 2015 | WW | WW | SM | WW | SM | WB |
| 2016 | GM | WR | WW | SB | SM | SM |
| 2017 | WW | WW | WW | SM | WB | WB |

WW-winter wheat, WR-winter rapeseed, SM-silage maize, GM-grain maize, SB-summer barley, WB-winter barley, SP-spelt.

**Table 3.** Annual mean energy balance closure (EBC, slope of linear regression) and energy balance ratio (EBR) at the eddy covariance stations EC1 to EC6 in Kraichgau and Swabian Jura during 2010 – 2017. Regressions are based on half-hourly data.

| Region | | | Kraichgau | | | Swabian Jura | | | |
|---|---|---|---|---|---|---|---|---|---|
| | | | Sites | | | Sites | | | |
| Growing season, year | Parameter | Unit | EC1 | EC2 | EC3 | EC4 | EC5 | EC6 | Mean |
| 2010 | Slope | | 0.82 | 0.69 | 0.70 | 0.87 | 0.74 | 0.74 | 0.76 |
| | Intercept | $W\,m^{-2}$ | -2.09 | -5.55 | 8.59 | 3.06 | 2.86 | 11.84 | 3.12 |
| | $R^2$ | | 0.91 | 0.85 | 0.84 | 0.94 | 0.86 | 0.90 | 0.88 |
| | EBR | | 0.80 | 0.66 | 0.77 | 0.89 | 0.75 | 0.79 | 0.78 |
| | *Residual* | $W\,m^{-2}$ | 23.5 | 62.7 | 30.8 | 22.8 | 45.7 | 43.5 | 38.2 |
| 2011 | Slope | | 0.70 | 0.76 | 0.70 | 0.77 | 0.77 | 0.72 | 0.74 |
| | Intercept | $W\,m^{-2}$ | -4.95 | -0.13 | 5.22 | 1.54 | 12.62 | 4.57 | 3.15 |
| | $R^2$ | | 0.95 | 0.94 | 0.86 | 0.92 | 0.88 | 0.94 | 0.92 |
| | EBR | | 0.69 | 0.76 | 0.73 | 0.78 | 0.82 | 0.74 | 0.75 |
| | *Residual* | $W\,m^{-2}$ | 54.8 | 32.1 | 55.4 | 52.4 | 43.0 | 63.7 | 50.2 |
| 2012 | Slope | | 0.74 | 0.67 | 0.69 | 0.81 | 0.78 | 0.72 | 0.74 |
| | Intercept | $W\,m^{-2}$ | -3.14 | 7.48 | 4.17 | 3.65 | 6.48 | 2.00 | 3.44 |
| | $R^2$ | | 0.96 | 0.86 | 0.94 | 0.90 | 0.89 | 0.93 | 0.91 |
| | EBR | | 0.73 | 0.69 | 0.71 | 0.84 | 0.82 | 0.74 | 0.76 |
| | *Residual* | $W\,m^{-2}$ | 53.2 | 75.6 | 57.3 | 24.7 | 31.5 | 38.4 | 46.8 |
| 2013 | Slope | | 0.66 | 0.71 | 0.70 | 0.79 | 0.67 | 0.72 | 0.71 |
| | Intercept | $W\,m^{-2}$ | -6.59 | -0.26 | 4.40 | 4.17 | -0.77 | 3.28 | 0.71 |
| | $R^2$ | | 0.95 | 0.96 | 0.95 | 0.92 | 0.94 | 0.93 | 0.94 |
| | EBR | | 0.62 | 0.71 | 0.72 | 0.82 | 0.67 | 0.74 | 0.71 |
| | *Residual* | $W\,m^{-2}$ | 59.8 | 42.1 | 53.9 | 32.5 | 52.8 | 46.4 | 48.0 |
| 2014 | Slope | | 0.69 | 0.74 | 0.70 | 0.79 | 0.66 | 0.74 | 0.72 |
| | Intercept | $W\,m^{-2}$ | 4.34 | 5.66 | 4.78 | -2.69 | 0.50 | 0.46 | 2.18 |
| | $R^2$ | | 0.89 | 0.86 | 0.92 | 0.93 | 0.93 | 0.94 | 0.91 |
| | EBR | | 0.71 | 0.77 | 0.73 | 0.78 | 0.66 | 0.75 | 0.73 |
| | *Residual* | $W\,m^{-2}$ | 51.2 | 36.7 | 39.2 | 43.7 | 69.9 | 44.9 | 47.6 |
| 2015 | Slope | | 0.71 | 0.81 | 0.77 | 0.81 | 0.73 | 0.76 | 0.77 |
| | Intercept | $W\,m^{-2}$ | -5.14 | -5.65 | 8.13 | -2.85 | 7.22 | -4.99 | -0.55 |
| | $R^2$ | | 0.96 | 0.94 | 0.92 | 0.94 | 0.87 | 0.94 | 0.93 |
| | EBR | | 0.67 | 0.77 | 0.81 | 0.79 | 0.77 | 0.72 | 0.76 |
| | *Residual* | $W\,m^{-2}$ | 46.4 | 29.1 | 34.4 | 30.4 | 42.9 | 36.9 | 36.7 |
| 2016 | Slope | | 0.75 | 0.83 | 0.80 | 0.77 | 0.62 | 0.75 | 0.75 |
| | Intercept | $W\,m^{-2}$ | 7.50 | -5.99 | 3.39 | -0.66 | 5.70 | 3.42 | 2.23 |
| | $R^2$ | | 0.89 | 0.92 | 0.94 | 0.92 | 0.84 | 0.88 | 0.90 |
| | EBR | | 0.79 | 0.78 | 0.82 | 0.76 | 0.66 | 0.79 | 0.77 |
| | *Residual* | $W\,m^{-2}$ | 40.8 | 23.3 | 33.3 | 28.8 | 47.9 | 22.3 | 32.7 |
| 2017 | Slope | | 0.70 | 0.73 | 0.86 | 0.84 | 0.66 | 0.76 | 0.76 |
| | Intercept | $W\,m^{-2}$ | -6.41 | -0.81 | 2.36 | 10.63 | 11.50 | -0.67 | 2.77 |
| | $R^2$ | | 0.96 | 0.95 | 0.93 | 0.86 | 0.87 | 0.94 | 0.92 |
| | EBR | | 0.66 | 0.73 | 0.87 | 0.90 | 0.72 | 0.75 | 0.77 |
| | *Residual* | $W\,m^{-2}$ | 48.2 | 52.8 | 17.4 | 20.5 | 58.4 | 39.7 | 32.9 |
| Mean | Slope | | 0.72 | 0.74 | 0.74 | 0.81 | 0.70 | 0.74 | 0.74 |
| | Intercept | $W\,m^{-2}$ | -2.06 | -0.66 | 5.13 | 2.11 | 5.76 | 2.49 | 2.13 |
| | $R^2$ | | 0.93 | 0.91 | 0.91 | 0.92 | 0.89 | 0.93 | 0.91 |
| | EBR | | 0.71 | 0.73 | 0.77 | 0.82 | 0.73 | 0.75 | 0.75 |
| | *Residual* | $W\,m^{-2}$ | 42.7 | 44.3 | 39.9 | 32.0 | 49.0 | 42.0 | 41.6 |




**Table 4.** Energy balance closure (EBC) as indicated by the slope of linear regression of turbulent vs. available energy and energy balance ratio (EBR) under different atmospheric stability conditions. EBC and EBR are given as site-specific averages from 2010 to 2017. SD: standard deviation.

| Region | | | Kraichgau | | | Swabian Jura | | |
|---|---|---|---|---|---|---|---|---|
| | | | Sites | | | Sites | | |
| Stability condition | Parameter | Unit | EC1 | EC2 | EC3 | EC4 | EC5 | EC6 |
| Unstable | Slope | | 0.69 | 0.74 | 0.70 | 0.79 | 0.68 | 0.74 |
| | Intercept | $W\,m^{-2}$ | -6.14 | -9.12 | 9.58 | 0.21 | 1.18 | -2.57 |
| | $R^2$ | | 0.87 | 0.85 | 0.83 | 0.87 | 0.84 | 0.8 |
| | EBR | | 0.67 | 0.69 | 0.73 | 0.78 | 0.68 | 0.73 |
| | SD(EBR) | | 0.32 | 0.34 | 0.37 | 0.28 | 0.32 | 0.27 |
| | *Residual* | $W\,m^{-2}$ | 94.7 | 82.0 | 81.4 | 59.1 | 95.1 | 78.4 |
| | SD (*Res*) | $W\,m^{-2}$ | 62.3 | 61.5 | 67.3 | 54.4 | 68.2 | 58.3 |
| | N* | | 5478 | 4992 | 5926 | 7145 | 5755 | 8533 |
| Neutral | Slope | | 0.73 | 0.79 | 0.75 | 0.82 | 0.75 | 0.77 |
| | Intercept | $W\,m^{-2}$ | -0.95 | 0.04 | 10.22 | 3.58 | 7.44 | 1.59 |
| | $R^2$ | | 0.90 | 0.88 | 0.87 | 0.89 | 0.85 | 0.89 |
| | EBR | | 0.72 | 0.78 | 0.73 | 0.85 | 0.78 | 0.78 |
| | SD (EBR) | | 0.63 | 0.67 | 0.64 | 0.83 | 0.70 | 0.65 |
| | *Residual* | $W\,m^{-2}$ | 61.5 | 43.3 | 48.2 | 34.37 | 52.1 | 45.6 |
| | SD(*Res*) | $W\,m^{-2}$ | 59.1 | 60.0 | 62.2 | 50.6 | 64.4 | 52.8 |
| | N* | | 9957 | 10781 | 11180 | 11575 | 9563 | 12149 |
| Stable | Slope | | 0.66 | 0.69 | 0.71 | 0.57 | 0.57 | 0.48 |
| | Intercept | $W\,m^{-2}$ | -3.25 | -1.88 | 6.48 | 4.06 | 6.17 | 3.01 |
| | $R^2$ | | 0.86 | 0.82 | 0.86 | 0.61 | 0.68 | 0.52 |
| | EBR | | 0.41 | 0.46 | 0.56 | 0.41 | 0.30 | 0.33 |
| | SD (*EB*R) | | 1.34 | 0.67 | 1.56 | 1.48 | 1.44 | 1.51 |
| | *Residual)* | $W\,m^{-2}$ | 18.7 | 13.3 | 6.4 | 7.3 | 6.23 | 6.6 |
| | SD(*Res*) | $W\,m^{-2}$ | 31.5 | 29.0 | 34.5 | 29.2 | 36.2 | 28.9 |
| | N* | | 1292 | 1398 | 1642 | 1105 | 787 | 1075 |

* Number of data points