# Peer review of "Evaluating multi-year, multi-site data on the energy balance closure of eddy-covariance flux measurements at cropland sites in southwest Germany"

_Biogeosciences, 2018_

## Referee Comment (RC1) · M. Zeri (Referee) · 10 Oct 2018

**General comments**

The article is well written and relevant to the topic of energy balance closure. The use of such a long-term dataset gives strength to the results and conclusions. The introduction is well written and covers all the issues and approaches regarding the energy balance closure. Site and instrumentation are properly described. From the methodology, it is clear that all appropriate methods and corrections to field data were

used.

I recommend minor revisions that included additional terms in the energy balance closure, or justification for not considering those terms. In addition, references to experiments and published results on advection should be included, since advection can have a large impact on measured fluxes over non-flat terrain.

Specific comments

Page 2, line 18: Replace "mesoscalic" with mesoscale

Page 7, line 12: Replace "instalment" with "set up" or remove it

Page 7, line 16: It would be better to rephrase "time variable canopy height" with something like: canopy height, which changed over time due to crop growth. Question here: how was the canopy height used in the flux software? Some flux software have the option to input heights over time. If this was the case in TK3, how frequently was the crop height changed in the software (bi-weekly, monthly. . .)?

Legend of Fig. 10: Space between "2017study"

Page 12, 25-30: If the storage due to photosynthesis is straightforward to calculate from CO2-fluxes, why was it not included in this study? How about the heat storage in the soil layer? It can be derived with measurements of soil temperature and soil moisture, which were available in the experimental setup described in this study. Both terms were included in the EBC for crops in this study:

Zeri, M.; Anderson-Teixeira, K.; Hickman, G.; Masters, M.; DeLucia, E.; Bernacchi, C. J. Carbon exchange by establishing biofuel crops in Central Illinois. Agric. Ecosyst. Environ. 2011, 144, 319–329, doi:10.1016/j.agee.2011.09.006.

It would greatly help if both terms were included for at least one year per site, to compare with the use of only Rn, G, H and LE.

Page 14, line 15: "Moreover, the sources of secondary circulations are unclear, and

they are most probably not well linked with the locally measured available energy"
Complex topography can induce advective fluxes of CO2 and energy (Feigenwinter et
al. 2008; Rebmann et al. 2010).

Rebmann, C.; Zeri, M.; Lasslop, G.; Mund, M.; Kolle, O.; Schulze, E.-D.; Feigen-
winter, C. Treatment and assessment of the CO2-exchange at a complex for-
est site in Thuringia, Germany. Agric. For. Meteorol. 2010, 150, 684–691,
doi:10.1016/j.agrformet.2009.11.001.

Feigenwinter, C.; Bernhofer, C.; Eichelmann, U.; Heinesch, B.; Hertel, M.; Janous,
D.; Kolle, O.; Lagergren, F.; Lindroth, A.; Minerbi, S.; Moderow, U.; Mölder, M.; Mon-
tagnani, L.; Queck, R.; Rebmann, C.; Vestin, P.; Yernaux, M.; Zeri, M.; Ziegler, W.;
Aubinet, M. Comparison of horizontal and vertical advective CO2 fluxes at three forest
sites. Agric. For. Meteorol. 2008, 148, 12–24, doi:10.1016/j.agrformet.2007.08.013.

Page 15, line 23: Another reference to katabatibc advection:

Heinesch, B.; Yernaux, Y.; Aubinet, M. Dependence of CO2 advection patterns
on wind direction on a gentle forested slope. Biogeosciences 2008, 5, 657–668,
doi:10.5194/bg-5-657-2008.

Page 15, line 25: Advective fluxes are mentioned here but not explained before or any
citation to experiments are given.

Page 16, References: Missing reference cited in the text:

Zeri, M.; Sá, L. D. A. The impact of data gaps and quality control filtering on the bal-
ances of energy and carbon for a Southwest Amazon forest. Agric. For. Meteorol.
2010, 150, 1543–1552, doi:10.1016/j.agrformet.2010.08.004.

---

## Referee Comment (RC2) · Anonymous Referee #2 · 31 Oct 2018

General Comments:

The manuscript is aimed to evaluate the Energy Balance Closure (EBC) on two different experimental sites by using multi-year datasets, and to assess how different factors (as topography, micrometeorological conditions, and different crops) may impact the EBC. The topic is of great interest because understanding the main factors that affect the EBC may have strong implications on the interpretation of energy flux measurements and on improving regional weather and global climate models.

[Figure]

The topic is appropriate for the publication on BioGeosciences. On the whole the paper is well structured and the obtained results are interesting. However, there are some important points that should be revised (specified in major comments). I recommend accepting the paper after having addressed the following major revisions:

Major Comments:

1) My first comment concerns the neglect of the 'minor (?) storage terms'. For example, ground heat flux is a significant component of the surface energy balance. Therefore, an accurate measurement of this term (or also of other terms that could be evaluated) is fundamental for improving the EBC. A lot of studies in literature demonstrated the importance of the correction for heat storage into the soil that greatly improved the global closure rate. I think the authors could easily account for some of these terms with the available data. They should try to first apply the possible corrections to evaluate the EBC as better as possible; subsequently, they can investigate in a more rigorous way the effect of the other factors (as topographical and micrometeorological characteristics).

2) Effect of the wind direction on EBC: Authors observed that the computed EBR was highest for the prevailing wind directions at all the measurement sites. However, in my opinion, it is not the wind direction that affects the EBR, but the wind speed associated to the main wind direction. As a matter of fact, by comparing Figures 3 and 8, it is evident that the main wind directions correspond to the highest values of the wind velocity. As a consequence, I would discuss more the effect of the wind intensities than the effect of the wind directions. My impression is that the EBR is lowest in low-wind conditions (associated to less frequent wind directions) because of a higher uncertainty in the estimation of the turbulent fluxes in these situations.

3) Effect of the atmospheric conditions on EBC: Authors chose three statistics in order to investigate the effect of the atmospheric boundary layer flow on the EBC. In particular, they chose the kinematic virtual temperature flux (w'Tv') as a proxy for buoyancy

and they observed that larger heat fluxes (in modulus in stable conditions) correspond to a better EBC. Therefore they concluded that 'strong buoyancy' (that they correlated to high values of the modulus of w'Tv') produce a better EBC. However, this deduction is misleading because the highest values of |w'Tv'| (namely the highest values of downward heat flux) are not related to the highest values of buoyancy, or to very strong stratification of the atmosphere. Recent studies (Acevedo et al., 2016; Lan et al., 2018) investigated the transition between the weakly and the very stable boundary layer and highlighted the different behavior of momentum and heat fluxes as stability increases. Whereas the momentum flux tends to progressively decrease as the stability increases, the heat flux increases in weakly stable conditions when the mechanical mixing weakens the magnitude of mean temperature gradient and allows turbulent eddies with larger vertical scales to develop. The magnitude of downward sensible heat flux is mainly dependent on the small vertical temperature gradient and the large turbulent heat diffusivity. The downward kinematic heat flux reaches a maximum value under 'moderately stable conditions' (the turning point). This stability turning point marks the transition from weakly to strongly stable regimes, when the weak mechanical mixing favors the buildup of strong stratifications, induced by the surface radiative cooling, which in turn confines turbulent eddies within thin layers locally. Such suppressed turbulent eddies are responsible for the limited downward heat flux that dramatically decreases in very stable conditions (after the turning point).

Therefore the observed low values of downward w'Tv' are not necessarily associated to transition periods between daytime and nightime conditions (as the authors claimed, cft text pag. 11, lines 7-9), but they could be related to periods of very stable conditions. These periods are usually associated to low winds and to weak level of turbulence interrupted by intermittent bursts often induced by submeso motions (Cava et al., 2015; 2016; Mortarini et al., 2018). The authors could check the wind intensity and the atmospheric stability correspondent to the low values of w'Tv'.

Summarizing, my impression is that the EBC improves in moderately stable conditions,

and worsen in very stable conditions due to the high uncertainty in the estimation of the very low turbulent fluxes related to the weak and intermittent character of turbulent flow.

4) Pag., 14 – Lines 6-9 'Our finding of the highest EBR at the two sites with the most pronounced buoyancy does not fit well with studies that recommended considering secondary circulations to achieve a better EBC (Cava et al., 2008; Foken et al., 2006; Kidston et al., 2010, Mauder et al. 2010). Those studies postulate that heterogeneity-induced and buoyancy-driven quasi-stationary circulations are probably the dominant processes behind underestimated energy fluxes.'

The studies that suggested the use of an averaging period higher that 30 minutes usually refer to unstable conditions. These studies suggested that averaging periods of 2–4 h are often needed to statistically resolve the largest convective turbulent eddies or also non-stationary mesoscale motions that sometimes can modulate turbulent fluxes (Mahrt, 1998). Differently, in the previous sentence the authors are discussing the behavior of EBR at EC2 and EC4 for negative (downward) heat fluxes (i.e. stable conditions). Cava et al. (2008) showed as the application of a larger averaging period improved the short term EBC during the diurnal hours, but not in stable conditions during the night. Therefore the previous sentence and the interpretation of results should be modified, accordingly to the previous comment.

5) Pag., 14 – Line 12 'Finding an optimum averaging period is a very complex to nearly impossible task. ' – Finding an optimum averaging period for computing turbulent statistics that holds for all the atmospheric conditions is impossible. The choice of the averaging period depends on the aim of the analysis and on the involved characteristic time scales. The classical averaging period of 30 minutes can be a proper choice for unstable or neutral conditions, even if, as already discussed, a larger period could be useful to better resolve larger scales that contribute to the transport in these conditions. On the other hand, the computation of turbulence statistics in very stable conditions requires the use of a shorter averaging time (few minutes, according to Sun et al.,

2012, Vickers and Mahrt (2006) or the various Mahrt's papers). Probably the use of a shorter time scale in stable conditions could improve also the EBC at the corresponding hours.

References: Acevedo, O. C., Mahrt, L., Puhales, F. S., Costa, F. D., Medeiros, L. E., & Degrazia, G. A. (2016) Contrasting structures between the decoupled and coupled states of the stable boundary layer. Quarterly Journal of the Royal Meteorological Society, 142(695), 693–702.

Cava, D., Giostra, U., & Katul, G. (2015). Characteristics of gravity waves over an antarctic ice sheet during an Austral summer. Atmosphere, 6(9), 1271–1289.

Cava, D., Mortarini, L., Giostra, U., Richiardone, R., & Anfossi, D. (2016). A wavelet analysis of low-wind-speed submeso motions in a nocturnal boundary layer. Quarterly Journal of the Royal Meteorological Society, 143(703), 661–669.

Lan, C., Liu, H., Li, D., Katul, G. G., & Finn, D. (2018) Distinct turbulence structures in stably stratified boundary layers with weak and strong surface shear. Journal of Geophysical Research: Atmospheres, 123, 7839–7854.

Mortarini, L., Cava, D., Giostra, U., Acevedo, O., Nogueira Martins, L. G., Soares de Oliveira, P. E., & Anfossi, D. (2018). Observations of submeso motions and intermittent turbulent mixing across a low level jet with a 132-m tower. Quarterly Journal of the Royal Meteorological Society, 144(710), 172–183.

Sun J, Mahrt L, Banta RM, Pichugina YL. 2012. Turbulence regimes and turbulence intermittency in the stable boundary layer during CASES-99. J. Atmos. Sci. 69: 338–351.

Vickers D, Mahrt L. 2006. A solution for flux contamination by mesoscale motions with very weak turbulence. Boundary-Layer Meteorol. 118: 431–447.

Minor Comments:

1) Abstract: Pag. 1 - Line 20: 'To investigate the reasons behind EBC more closely for agro-ecosystems, ....' – This sentence is not clear; please, rephrase. 2) Abstract: Pag. 1 - Line 31: 'The measurement site exerted a statistically significant effect on EBC, but not crop or region' – What does it mean that the 'measurement site affect the EBC, but not the 'region'?. I cannot understand the difference. Please, better explain. 3) Pag. 7 – Lines 16 - 17 'Data for footprint analyses were constrained to u* > 0.1 m s−1 and $\zeta \geq -15.5$.' What is the motivation of the choice (-15.5) as a threshold for stability? 4) Pag. 10 – lines 16 - 17 : 'The statistical analyses showed that the EBC did not differ between the two regions (Fig. 7a) over the main vegetation period from April to June.' Why from April to June? Are the statistics shown in Figure 7 relative to all data sets or are restricted only to two months each year? If this is the case, please, motivate this choice. 5) Pag. 10 – lines 17 -18: 'The EBC measured at stations EC2 and EC4 was significantly higher (p < 0.001) than ....' - What is 'p'? I missed its definition in the text. 6) Pag. 13 – Line 2: 'In both KR and SJ, EBR was highest for winds blowing from the prevailing wind direction'- This is due to the higher wind speeds, as already discussed (see major comment 2). 7) Pag. 13 – Lines 6-15: This discussion should be inserted in a section relative to the effect of the instrumental setup ... not in this section (Effect of atmospheric conditions on EBC)! 8) Pag. 13 – Line 18: 'Their results confirm that their EC site had various turbulence and closure patterns'. Please, rephrase the sentence because it is unclear. 9) Pag. 14 – Lines 4-5: 'At these two sites, strong negative buoyancy fluxes below −0.15 K m s−1 were recorded. This means that the atmosphere was not heated by the land surface, but that the land surface was significantly heated by the atmosphere.' Probably, the authors would like to say that in stable atmosphere there is a downward heat transfer? I cannot understand the motivation of this sentence and its connection with the next sentence (see major comment (4)). Please, rephrase (or cut) the sentence because it is unclear. 10) References: Please, pay attention to the references because some papers are cited in the text, but are missing in the list.

Please also note the supplement to this comment:
https://www.biogeosciences-discuss.net/bg-2018-422/bg-2018-422-RC2-
supplement.pdf

---

## Author Comment (AC1) · 6 Dec 2018

**Response to reviewers comments (bg-2018-422)**

We would like to thank the editor for handling our manuscript and finding two constructive reviewers. Additionally, we wish to thank the editor for extending the deadline which ensured we could properly complete the revision of the manuscript. We also would like to thank the two reviewers for their careful and thorough reading of this manuscript and for the thoughtful comments and constructive suggestions. This has helped us to further improve the quality of this manuscript. Moreover, we have taken the trouble to enhance readability at some few selected places throughout the revised manuscript, marked in green font, the intended meaning has been maintained.

Our response follows (the reviewer's comments and our responses in blue are given below. Changes to the text and citations from the have additionally been marked in italics.)

**Response to Reviewer #1 (bg-2018-422-RC1)**

Specific comments

Page 2, line 18: Replace "mesoscalic" with mesoscale
Rephrased as suggested.

Page 7, line 12: Replace "instalment" with "set up" or remove it
Thanks for this remark. The word "*instalment*" was deleted.

Page 7, line 16: It would be better to rephrase "time variable canopy height" with something like: canopy height, which changed over time due to crop growth. Question here: how was the canopy height used in the flux software? Some flux software have the option to input heights over time. If this was the case in TK3, how frequently was the crop height changed in the software (bi-weekly, monthly: : :)?

Correct. The sentence was rephrased and now reads as follows (page 7, line 16): *"[…] canopy height, which changed over time due to crop growth, was measured biweekly. The mean measured crop height was considered in TK3 for the respective two-week periods. "*

Legend of Fig. 10: Space between "2017study"

Sorry, that was a typo. Corrected.

Page 12, 25-30: If the storage due to photosynthesis is straightforward to calculate from $CO_2$-fluxes, why was it not included in this study? How about the heat storage in the soil layer? It can be derived with measurements of soil temperature and soil moisture, which were available in the experimental setup described in this study. Both terms were included in the EBC for crops in this study: Zeri, M.; Anderson-Teixeira, K.; Hickman, G.; Masters, M.; DeLucia, E.; Bernacchi, C.J. Carbon exchange by establishing biofuel crops in Central Illinois. Agric. Ecosyst. Environ. 2011, 144, 319–329, doi:10.1016/j.agee.2011.09.006.
It would greatly help if both terms were included for at least one year per site, to compare with the use of only Rn, G, H and LE.

The soil heat storage was considered in hour study. On p. 5, line 6-7, we write *"...Data from thermistor (0.02 m and 0.06 m) and FDR sensors (0.05 m) were used to calculate the soil heat storage between the soil heat flux plates and the ground surface ..."*. We have now added references to Wizemann et al. (2014) and Eshonkulov et al. (2019), were this has already been calculated and discussed in great detail.

In the previous study (Eshonkulov et al., 2019) had quantified minor storage terms and assessed their effect on the EBC. There it was found that all minor storage terms (enthalpy change in the plant canopy, the air enthalpy change, the energy consumption and release by photosynthesis and respiration, and the atmospheric moisture change) together increased EBC by 5% to 6.8% on average. Among the terms, energy consumption and release by photosynthesis and respiration dominated with an increase of EBC between 4.7% and 5.1%. We discuss this issue on p. 12, line 20-27.

Page 14, line 15: "Moreover, the sources of secondary circulations are unclear, and they are most probably not well linked with the locally measured available energy". Complex topography can induce advective fluxes of $CO_2$ and energy (Feigenwinter et al. 2008; Rebmann et al. 2010).

Rebmann, C.; Zeri, M.; Lasslop, G.; Mund, M.; Kolle, O.; Schulze, E.-D.; Feigenwinter, C. Treatment and assessment of the $CO_2$-exchange at a complex forest site in Thuringia, Germany. Agric. For. Meteorol. 2010, 150, 684–691, doi:10.1016/j.agrformet.2009.11.001.

Feigenwinter, C.; Bernhofer, C.; Eichelmann, U.; Heinesch, B.; Hertel, M.; Janous, D.; Kolle, O.; Lagergren, F.; Lindroth, A.; Minerbi, S.; Moderow, U.; Mölder, M.; Montagnani, L.; Queck, R.; Rebmann, C.; Vestin, P.; Yernaux, M.; Zeri, M.; Ziegler, W.; Aubinet, M. Comparison of horizontal and vertical advective $CO_2$ fluxes at three forest sites. Agric. For. Meteorol. 2008, 148, 12–24, doi:10.1016/j.agrformet.2007.08.013.

We thank the reviewer for this remark. We follow the reviewer's suggestion by rephrasing to (now on page 16, line 4-5):

 "Moreover, *complex topography can induce advective fluxes (Feigenwinter et al., 2008; Rebmann et al., 2010). Therefore,* the former waste dump located about […].

The references have additionally been added to the reference list.

Page 15, line 23: Another reference to katabatic advection: Heinesch, B.; Yernaux, Y.; Aubinet, M. Dependence of $CO_2$ advection patterns on wind direction on a gentle forested slope. Biogeosciences 2008, 5, 657–668, doi:10.5194/bg-5-657-2008.

Many thanks for suggesting the additional reference to katabatic advection which we have now included on page 16, line 3 and added to the reference list.

Page 15, line 25: Advective fluxes are mentioned here but not explained before or any citation to experiments are given.

This point is covered by responding to the two previous comments, and adding the references stated, there.

Page 16, References: Missing reference cited in the text:
Zeri, M.; Sá, L. D. A. The impact of data gaps and quality control filtering on the balances of energy and carbon for a Southwest Amazon forest. Agric. For. Meteorol. 2010, 150, 1543–1552, doi:10.1016/j.agrformet.2010.08.004.

Well spotted. We now added the paper to the list of references. (Page 22, Line: 34-35)

**Additional changes:**

1)
The work was in parts supported by a previously not listed source of funding. Therefore, we added the following sentence to the acknowledgements.: "*Additionally, this work received support from the funding by the Collaborative Research Center 1253 CAMPOS (Project 7: Stochastic Modelling Framework), funded by the German Research Foundation (DFG, Grant Agreement SFB 1253/1 2017).*"

2)
To enhance the flow of the text, we have made a slight change to the introduction, explained in the following:

The paragraph of the original manuscript on page 3 lines 7-16 was moved to what is now page 2 lines 29-27 and the first sentence was deleted. The moved paragraph has been marked in green in the revised manuscript.

3)
The reference to Eshonkulov et al. (2018) has now been updated to Eshonkulov et al. (2019), since in the meantime it has been published.

4)
We deleted the sentence: "… *Note, however, that the difference of residual energy under stable conditions may be the result of using only daytime data (from 7 am to 7 pm)…*" because it was not well connected to the previous part.

5)
Because the sentence was misleading we rephrased the sentence "…*Eshonkulov et al. (2019) demonstrated that minor storage and flux terms over winter wheat in southwest Germany contributed the most to the EBC during the main vegetation period in May…*" into "…*Eshonkulov et al. (2019) demonstrated that the contribution of minor storage and flux terms over winter wheat in southwest Germany was largest during the main vegetation period in May…*"

6)
We extended the acknowledgements to the associate editor and the two involved reviewers.
"*We thank Dr. Paul Stoy for handling the manuscript, one anonymous reviewer and Marcelo Zeri for helpful and constructive comments.*"

7)
To enhance readability, we rephrased the sentence now on page 13 line 16 to read "*At our study sites, neutral conditions dominated (~ 60 %), followed by unstable conditions (~ 34 %) and by stable conditions (6 %) (Table 4)*"

8)

The email of corresponding author was changed to ravshan.eshonkulov@qmii.uz.

---

## Author Comment (AC2) · 6 Dec 2018

**Response to reviewers comments (bg-2018-422)**

We would like to thank the editor for handling our manuscript and finding two constructive reviewers. Additionally, we wish to thank the editor for extending the deadline which ensured we could properly complete the revision of the manuscript. We also would like to thank the two reviewers for their careful and thorough reading of this manuscript and for the thoughtful comments and constructive suggestions. This has helped us to further improve the quality of this manuscript. Moreover, we have taken the trouble to enhance readability at some few selected places throughout the revised manuscript, marked in green font, the intended meaning has been maintained.

Our response follows (the reviewer's comments and our responses in blue are given below. Changes to the text and citations from the have additionally been marked in italics.)

**Response to Reviewer #2 (bg-2018-422-RC2)**

**Major comments:**

1) My first comment concerns the neglect of the 'minor (?) storage terms'. For example, ground heat flux is a significant component of the surface energy balance. Therefore, an accurate measurement of this term (or also of other terms that could be evaluated) is fundamental for improving the EBC. A lot of studies in literature demonstrated the importance of the correction for heat storage into the soil that greatly improved the global closure rate. I think the authors could easily account for some of these terms with the available data. They should try to first apply the possible corrections to evaluate the EBC as better as possible; subsequently, they can investigate in a more rigorous way the effect of the other factors (as topographical and micrometeorological characteristics).

While the reviewer is of course correct in making this statement. Nevertheless, there is a misunderstanding because the soil heat storage was both **measured** and **considered** in our study! On p. 5, line 2-9, we write *"…To measure the soil heat flux near the EC stations, three heat flux plates (HFP01, Hukseflux Thermal sensors, Delft, The Netherlands) were installed at a depth of 0.08 m … Data from thermistor (0.02 m and 0.06 m) and FDR sensors (0.05 m) were used to calculate the soil heat storage between the soil heat flux plates and the ground surface. …".* The ground heat flux G given in Eq. 1-3 is the soil heat flux in 0.08 m plus the heat storage change in the layer above the plates.

We have now added references to Wizemann et al. (2014) and Eshonkulov et al. (2019). In these papers, the calculation is described in detail.

In a previous study we quantified minor storage terms and assessed their effect on the EBC (Eshonkulov et al., 2019). There we found that all minor storage terms (enthalpy change in the plant canopy, the air enthalpy change, the energy consumption and release by photosynthesis and respiration, and the atmospheric moisture change) together increased EBC by 5% to 6.8% on average. Among the terms, energy consumption and release by photosynthesis and respiration dominated with an increase of EBC between 4.7% and 5.1%. We discuss this issue on p. 12, line 20-27.

2) Effect of the wind direction on EBC: Authors observed that the computed EBR was highest for the prevailing wind directions at all the measurement sites. However, in my opinion, it is not the wind

direction that affects the EBR, but the wind speed associated to the main wind direction. As a matter of fact, by comparing Figures 3 and 8, it is evident that the main wind directions correspond to the highest values of the wind velocity. As a consequence, I would discuss more the effect of the wind intensities than the effect of the wind directions. My impression is that the EBR is lowest in low-wind conditions (associated to less frequent wind directions) because of a higher uncertainty in the estimation of the turbulent fluxes in these situations.

We thank the reviewer for this valuable comment. We renamed section 3.4.2 (Page 10, Line 13) to '*Effect of wind speed and direction*' and moved the paragraphs discussing the horizontal wind speeds from '*Meteorological and terrain conditions*' to this section (Page 10, Line 14-23).

3) Effect of the atmospheric conditions on EBC: Authors chose three statistics in order to investigate the effect of the atmospheric boundary layer flow on the EBC. In particular, they chose the kinematic virtual temperature flux (w'Tv') as a proxy for buoyancy and they observed that larger heat fluxes (in modulus in stable conditions) correspond to a better EBC. Therefore they concluded that 'strong buoyancy' (that they correlated to high values of the modulus of w'Tv') produce a better EBC. However, this deduction is misleading because the highest values of |w'Tv'| (namely the highest values of downward heat flux) are not related to the highest values of buoyancy, or to very strong stratification of the atmosphere. Recent studies (Acevedo et al., 2016; Lan et al., 2018) investigated the transition between the weakly and the very stable boundary layer and highlighted the different behavior of momentum and heat fluxes as stability increases. Whereas the momentum flux tends to progressively decrease as the stability increases, the heat flux increases in weakly stable conditions when the mechanical mixing weakens the magnitude of mean temperature gradient and allows turbulent eddies with larger vertical scales to develop. The magnitude of downward sensible heat flux is mainly dependent on the small vertical temperature gradient and the large turbulent heat diffusivity. The downward kinematic heat flux reaches a maximum value under 'moderately stable conditions' (the turning point). This stability turning point marks the transition from weakly to strongly stable regimes, when the weak mechanical mixing favors the buildup of strong stratifications, induced by the surface radiative cooling, which in turn confines turbulent eddies within thin layers locally. Such suppressed turbulent eddies are responsible for the limited downward heat flux that dramatically decreases in very stable conditions (after the turning point). Therefore the observed low values of downward w'Tv' are not necessarily associated to transition periods between daytime and nightime conditions (as the authors claimed, cft text pag. 11, lines 7-9), but they could be related to periods of very stable conditions. These periods are usually associated to low winds and to weak level of turbulence interrupted by intermittent bursts often induced by submeso motions (Cava et al., 2015; 2016; Mortarini et al., 2018). The authors could check the wind intensity and the atmospheric stability correspondent to the low values of w'Tv'. Summarizing, my impression is that the EBC improves in moderately stable conditions, and worsen in very stable conditions due to the high uncertainty in the estimation of the very low turbulent fluxes related to the weak and intermittent character of turbulent flow.

We really appreciate this comment. Particularly the paper of Lan et al. (2018) was very helpful in the interpretation of our results. We agree with the reviewer's point and add the following (p. 13, Line 24-35; changes in italics): *"… The scatter of EBR versus buoyancy flux at EC2 and EC4, the two stations with the highest EBC, differed from those of the other stations (Fig. 10). At these two sites, strong negative buoyancy fluxes below −0.15 K m s⁻¹ were recorded. This means that the atmosphere was not heated by the land surface, but that the land surface was significantly heated by the atmosphere. Such a situation points to a stable boundary layer (SBL). Lan et al. (2018) report that they measured the highest buoyancy fluxes under a weak SBL with strong surface shear. They argue that the strong mechanical shear produced at the ground favors the development of turbulent eddies with larger scales that enhance*

*vertical mixing of momentum and heat transporting the aloft warm air downward and the surface cold air upward. Moreover, the mechanical mixing weakens the magnitude of mean temperature gradient and allows turbulent eddies with larger vertical scales to develop. Conversely, under a strongly SBL weak winds occur near the surface and turbulent eddies are depressed and detached from the boundary leading to suppressed vertical mixing. Several studies recommended considering secondary circulations to achieve a better EBC (Foken et al., 2010; Kidston et al., 2010; Mauder et al., 2010).".*
We added the reference to Lan et al. (2018) to the reference list.

4) Pag., 14 – Lines 6-9 'Our finding of the highest EBR at the two sites with the most pronounced buoyancy does not fit well with studies that recommended considering secondary circulations to achieve a better EBC (Cava et al., 2008; Foken et al., 2006; Kidston et al., 2010, Mauder et al. 2010). Those studies postulate that heterogeneity induced and buoyancy-driven quasi-stationary circulations are probably the dominant processes behind underestimated energy fluxes. The studies that suggested the use of an averaging period higher that 30 minutes usually refer to unstable conditions. These studies suggested that averaging periods of 2–4 h are often needed to statistically resolve the largest convective turbulent eddies or also non-stationary mesoscale motions that sometimes can modulate turbulent fluxes (Mahrt, 1998). Differently, in the previous sentence the authors are discussing the behavior of EBR at EC2 and EC4 for negative (downward) heat fluxes (i.e. stable conditions). Cava et al. (2008) showed as the application of a larger averaging period improved the short term EBC during the diurnal hours, but not in stable conditions during the night. Therefore the previous sentence and the interpretation of results should be modified, accordingly to the previous comment.

We agree and have removed the speculation that the high EBR at EC2 and EC4 is related to pronounced buoyancy. Now, we simply state that several studies recommended considering secondary circulations to achieve a better EBC. p. 14, line 1-2: *"…Several studies recommended considering secondary circulations to achieve a better EBC (Foken et al., 2010; Kidston et al., 2010; Mauder et al., 2010). Those studies postulate that heterogeneity-induced and buoyancy-driven quasi-stationary circulations are probably the dominant processes behind underestimated energy fluxes…"*

Thank you for this suggestion. The sentences modified (in discussion part) and the interpretation of the results was changed (Page 14, Line: 6-7, 10-13).

5) Pag., 14 – Line 12 'Finding an optimum averaging period is a very complex to nearly impossible task. – Finding an optimum averaging period for computing turbulent statistics that holds for all the atmospheric conditions is impossible. The choice of the averaging period depends on the aim of the analysis and on the involved characteristic time scales. The classical averaging period of 30 minutes can be a proper choice for unstable or neutral conditions, even if, as already discussed, a larger period could be useful to better resolve larger scales that contribute to the transport in these conditions. On the other hand, the computation of turbulence statistics in very stable conditions requires the use of a shorter averaging time (few minutes, according to Sun et al., 2012, Vickers and Mahrt (2006) or the various Mahrt's papers). Probably the use of a shorter time scale in stable conditions could improve also the EBC at the corresponding hours.

You are right. Therefore, the sentence has been modified and now reads (Page 14, Line: 10-13): *"The classical averaging period of 30 minutes can be a proper choice for unstable or neutral conditions. Shorter averaging period is suitable for capturing energy fluxes in very stable conditions [...].*

References: Acevedo, O. C., Mahrt, L., Puhales, F. S., Costa, F. D., Medeiros, L. E., & Degrazia, G. A. (2016) Contrasting structures between the decoupled and coupled states of the stable boundary layer. Quarterly Journal of the Royal Meteorological Society, 142(695), 693–702.

Cava, D., Giostra, U., & Katul, G. (2015). Characteristics of gravity waves over an antarctic ice sheet during an Austral summer. Atmosphere, 6(9), 1271–1289.

Cava, D., Mortarini, L., Giostra, U., Richiardone, R., & Anfossi, D. (2016). A wavelet analysis of low-wind-speed submeso motions in a nocturnal boundary layer. Quarterly Journal of the Royal Meteorological Society, 143(703), 661–669.

Lan, C., Liu, H., Li, D., Katul, G. G., & Finn, D. (2018) Distinct turbulence structures in stably stratified boundary layers with weak and strong surface shear. Journal of Geophysical Research: Atmospheres, 123, 7839–7854.

Mortarini, L., Cava, D., Giostra, U., Acevedo, O., Nogueira Martins, L. G., Soares de Oliveira, P. E., & Anfossi, D. (2018). Observations of submeso motions and intermittent turbulent mixing across a low level jet with a 132-m tower. Quarterly Journal of the Royal Meteorological Society, 144(710), 172–183.

Sun J, Mahrt L, Banta RM, Pichugina YL. 2012. Turbulence regimes and turbulence intermittency in the stable boundary layer during CASES-99. J. Atmos. Sci. 69: 338–351.

Vickers D, Mahrt L. 2006. A solution for flux contamination by mesoscale motions with very weak turbulence. Boundary-Layer Meteorol. 118: 431–447

**Minor Comments:**

1) Abstract: Pag. 1 - Line 20: 'To investigate the reasons behind EBC more closely for agro-ecosystems, ...' – This sentence is not clear; please, rephrase.

The sentence was rephrased as follows (Page. 1, Line: 20-21): *"To investigate the nature of the energy balance gap for agro-ecosystems, […]"*.

2) Abstract: Pag. 1 - Line 31: 'The measurement site exerted a statistically significant effect on EBC, but not crop or region' – What does it mean that the 'measurement site affect the EBC, but not the 'region'?. I cannot understand the difference. Please, better explain.

We replaced the term *"measurement site"* with the clearer term *"study site"* and rephrased the sentence as follows *(Page. 1, Line: 31)*: *"… The study site exerted a statistically significant effect on EBC but neither did crop and nor region (KR vs SJ)."*

3) Pag. 7 – Lines 16 - 17 'Data for footprint analyses were constrained to $u* > 0.1$ m s$^{-1}$ and $\zeta \geq 15.5$.' What is the motivation of the choice (-15.5) as a threshold for stability?

This threshold for stability was taken from Kljun et al., 2015 (Kljun, N., Calanca, P., Rotach, M. W. and Schmid, H. P.: A simple two-dimensional parameterisation for Flux Footprint Prediction (FFP), Geosci. Model Dev, 8, 3695–3713, doi:10.5194/gmd-8-3695-2015, 2015.) and regarding the refence that the footprint parametrisation is restricted to $-15.5 \leq z/L$ and $u*>0.1$ m s$^{-1}$ due to the height of the measurement and requirement of stationarity (page 3705, section 6.2, in cited reference).

4) Pag. 10 – lines 16 - 17: 'The statistical analyses showed that the EBC did not differ between the two regions (Fig. 7a) over the main vegetation period from April to June.' Why from April to June? Are the statistics shown in Figure 7 relative to all data sets or are restricted only to two months each year? If this is the case, please, motivate this choice.

Correct. The sentence has been changed to (changes in italics, page 10, line 8): "The statistical analyses showed that the EBC did not differ between the two regions (Fig. 6a) over the main vegetation period from *early April until harvest.* "

5) Pag. 10 – lines 17 -18: 'The EBC measured at stations EC2 and EC4 was significantly higher (p < 0.001) than …' - What is 'p'? I missed its definition in the text.

Correct. The p-value gives the level of significance or probability error. It is defined at first occurrence. The sentence now reads (page 10 line 8-9; changes in italics): "The EBC measured at stations EC2 and EC4 was significantly higher ($p < 0.001$; $p - probability\ level$) than at the other stations (Fig. 6b)".

6) Pag. 13 – Line 2: 'In both KR and SJ, EBR was highest for winds blowing from the prevailing wind direction'- This is due to the higher wind speeds, as already discussed (see major comment 2).

The sentence was changed to '*In both KR and SJ, EBR was highest for winds blowing from the prevailing wind direction. These winds were associated with high wind speeds favoring well-developed turbulent conditions*'. (now on Page 12, Lines: 30-31).

7) Pag. 13 –Lines 6-15: This discussion should be inserted in a section relative to the effect of the instrumental setup ... not in this section (Effect of atmospheric conditions on EBC)!

Yes, therefore, a new section was added titled: '*4.3 The effect of the instrumental setup*' and the paragraph pointed out by the reviewer moved to this section. (Page 15. Line: 3-12).

8) Pag. 13 – Line 18: 'Their results confirm that their EC site had various turbulence and closure patterns'. Please, rephrase the sentence because it is unclear.

Correct! And since there is no need for this sentence, we chose to delete it.

9) Pag. 14 –Lines 4-5: 'At these two sites, strong negative buoyancy fluxes below -0.15 K m s$^{-1}$ were recorded. This means that the atmosphere was not heated by the land surface, but that the land surface was significantly heated by the atmosphere.' Probably, the authors would like to say that in stable atmosphere there is a downward heat transfer? I cannot understand the motivation of this sentence and its connection with the next sentence (see major comment (4)). Please, rephrase (or cut) the sentence because it is unclear.

The sentence was rephrased see response to reviewer #2 comment 3).

10) References: Please, pay attention to the references because some papers are cited in the text, but are missing in the list.

Thanks for this remark. The missing references were inserted.

**Additional changes:**

1)
The work was in parts supported by a previously not listed source of funding. Therefore, we added the following sentence to the acknowledgements.: "*Additionally, this work received support from the funding by the Collaborative Research Center 1253 CAMPOS (Project 7: Stochastic Modelling Framework), funded by the German Research Foundation (DFG, Grant Agreement SFB 1253/1 2017)."*

2)
To enhance the flow of the text, we have made a slight change to the introduction, explained in the following:

The paragraph of the original manuscript on page 3 lines 7-16 was moved to what is now page 2 lines 29-27 and the first sentence was deleted. The moved paragraph has been marked in green in the revised manuscript.

3)
The reference to Eshonkulov et al. (2018) has now been updated to Eshonkulov et al. (2019), since in the meantime it has been published.

4)
We deleted the sentence: "*… Note, however, that the difference of residual energy under stable conditions may be the result of using only daytime data (from 7 am to 7 pm)…*" because it was not well connected to the previous part.

5)
Because the sentence was misleading we rephrased the sentence "*…Eshonkulov et al. (2019) demonstrated that minor storage and flux terms over winter wheat in southwest Germany contributed the most to the EBC during the main vegetation period in May…*" into "*…Eshonkulov et al. (2019) demonstrated that the contribution of minor storage and flux terms over winter wheat in southwest Germany was largest during the main vegetation period in May…*"

 6)
We extended the acknowledgements to the associate editor and the two involved reviewers.
"*We thank Dr. Paul Stoy for handling the manuscript, one anonymous reviewer and Marcelo Zeri for helpful and constructive comments.*"

7)
To enhance readability, we rephrased the sentence now on page 13 line 16 to read "*At our study sites, neutral conditions dominated (~ 60 %), followed by unstable conditions (~ 34 %) and by stable conditions (6 %) (Table 4)*"
8)
The email of corresponding author was changed to ravshan.eshonkulov@qmii.uz.

---

## Author Response (AR1)

**Response to editor comments (bg-2018-422)**

Dear Dr. Stoy,

thank you for your careful and thorough reading of our manuscript, in particular your constructive comments. In response to the minor concerns issued, we give some details below (in blue).

**Concerns addressed**

I feel that the manuscript could benefit strongly from a careful critique to improve flow.

And we strongly agree. Therefore, we have followed your suggestion. Thus, we have made quite some changes to the syntax of our manuscript, so that we now believe the flow has been greatly enhanced. Moreover, we have taken utmost (!) care to preserve the sense and meaning of our scientific results.

Page 2, line 18: Consider ' Across the world, research is being conducted to understand the reasons for the energy imbalance. One of the most extensive global EC networks is FLUXNET, with more than 500 EC towers around the world'. Across/around the world is mentioned twice in one sentence.

Thanks for this remark. The sentence was rephrased (Page 2, line 10-11). Now it reads as "Globally, a large number of research sites has been established to, inter alia, study reasons for the energy imbalance. This includes the FLUXNET network with more than 500 EC towers around the world (Wilson et al., 2002), and the AmeriFlux network operating in North, Central and South America (Peng et al., 2017)." (Page 2, Line: 1-13).

'A former waste dump' in the Figure 1 legend.

It does sound too colloquial. We changed the sentence to: *The yellow line demarks the boundaries of a former landfill site*.

Red and green should not be used simultaneously in Fig. 4 for our colorblind colleagues. (Fig. 7 is ok because green doesn't meaningfully enter that figure.)

Point well taken. The color of figure 4 was changed.

Take a close look at figure numbers: these repeat.

Repeated figure numbers were deleted.